# ProofOptimizer: Training Language Models to Simplify Proofs without Human Demonstrations

**Alex Gu**[†]
MIT CSAIL
gua@mit.edu

**Bartosz Piotrowski**[†]
Axiom Math
bartoszpiotrowski@post.pl

**Fabian Gloeckle**
Meta FAIR & Ecole des Ponts Paris
fgloeckle@meta.com

**Kaiyu Yang**[†]
kaiyuy@alumni.princeton.edu

**Aram H. Markosyan**[†]
Axiom Math
am@axiommath.ai

## Abstract

Neural theorem proving has advanced rapidly in the past year, reaching IMO gold-medalist capabilities and producing formal proofs that span thousands of lines. Although such proofs are mechanically verified by formal systems like Lean, their excessive length renders them difficult for humans to comprehend and limits their usefulness for mathematical insight. Proof simplification is therefore a critical bottleneck. Yet, training data for this task is scarce, and existing methods—mainly agentic scaffolding with off-the-shelf LLMs—struggle with the extremely long proofs generated by RL-trained provers. We introduce *ProofOptimizer*, the first language model trained to simplify Lean proofs without requiring additional human supervision. ProofOptimizer is trained via expert iteration and reinforcement learning, using Lean to verify simplifications and provide training signal. At inference time, it operates within an iterative proof-shortening workflow, progressively reducing proof length. Experiments show that ProofOptimizer substantially compresses proofs generated by state-of-the-art RL-trained provers on standard benchmarks, reducing proof length by 87% on miniF2F, 57% on PutnamBench, and 50% on Seed-Prover's IMO 2025 proofs. Beyond conciseness, the simplified proofs check faster in Lean and further improve downstream prover performance when reused as training data for supervised finetuning. We release our evaluation dataset and proofs generated by our model here.

## 1 Introduction

Theorem proving in formal environments such as Lean (de Moura et al., 2015) provides an excellent testbed for training large language models (LLMs) in mathematical reasoning via reinforcement learning (RL). Since Lean can mechanically verify proofs, it filters hallucinations and provides reliable reward signals, and enables enables unlimited high-quality synthetic reasoning data. Leveraging these benefits, LLMs finetuned with RL have achieved near gold-medal performance on the International Mathematical Olympiad (IMO) (Chen et al., 2025) and shown strong results on difficult college-level benchmarks like PutnamBench (Lin et al., 2025b).

However, RL-trained provers often generate proofs that are correct but excessively long and inscrutable. Since their only reward signal is the *correctness of generated proofs*, the resulting models produce proofs that are *correct* yet *suboptimal*: convoluted, bloated with redundant steps, or reliant on unnecessarily strong automation where a simple step would suffice. For example, Seed-Prover (Chen et al., 2025)'s Lean proof of IMO 2025 P1 consists of 4,357 lines of code, 16x longer (by character count) than its informal counterpart. Such proofs pose several practical drawbacks: they are (1) difficult for humans to comprehend, limiting their value as a source of mathematical insight; (2) less

---

[†]Work done at Meta Superintelligence Labs (FAIR).

suitable as synthetic training data, since models may struggle to learn from convoluted proofs; and (3) computationally inefficient to compile in Lean, which is especially problematic when integrated into existing formal libraries like mathlib (mathlib Community, 2019).

These challenges highlight the need for *proof simplification: transforming existing formal proofs into simpler forms while preserving correctness*. In this work, we adopt a natural notion of simplicity: *proof length*, measured by the number of Lean tokens. However, our approach is agnostic to the choice of simplicity metric: it is not restricted to proof length, but applies to any automatically computable measure (Kinyon, 2018).

Prior work on proof simplification (Ahuja et al., 2024) focuses on agentic scaffolding around API-only LLMs such as GPT-4o. While these methods can shorten human-written Lean proofs, they are ineffective at simplifying the long proofs generated by SoTA RL-trained LLM provers such as Seed-Prover and Goedel-Prover-V2 (Lin et al., 2025b), precisely the setting where simplification is most valuable. A natural alternative is to finetune LLMs directly for proof simplification, but progress in this direction is limited by the lack of suitable training data, namely aligned pairs of proofs before and after simplification.

We introduce *ProofOptimizer*, an LLM-based system for simplifying long and convoluted proofs in Lean. ProofOptimizer integrates three components: (i) a symbolic Lean linter that identifies and removes redundant steps, (ii) a 7B parameter language model finetuned specifically for proof simplification, and (iii) an iterative inference-time algorithm for progressively shortening proofs. Given an input proof, the Lean linter first eliminates the most obvious redundancies. The language model then generates multiple candidate simplifications, and the iterative algorithm repeatedly applies the model to the currently shortest proof, further reducing its length. Training follows two paradigms. In expert iteration, the model proposes simplifications that are verified by Lean and incorporated into the training data for supervised finetuning. In reinforcement learning, proof length and correctness serve as the reward signal. Both approaches enable continual improvement without requiring any human-annotated simplification data.

First, we evaluate ProofOptimizer on long proofs generated by state-of-the-art neural theorem provers. Specifically, we consider proofs produced by Goedel-Prover-V2 on two standard benchmarks— MiniF2F (Zheng et al., 2021) and PutnamBench—as well as four proofs released by Seed-Prover for IMO 2025. Our final models achieve significant results (Fig. 1), shortening MiniF2F proofs by an average of 63% in a single shot and PutnamBench proofs by 26% with 32 attempts, substantially outperforming Gemini-2.5-Pro (Sec. 4.1). At inference time, test-time RL improves single-shot miniF2F performance to 72%. With with iterative shortening, we achieve further per-proof average reductions of 87% (MiniF2F) and 57% (PutnamBench) and reduce the length of three out of four Seed-Prover IMO 2025 proofs by more than half.

Second, we conduct ablation studies to evaluate the effect of key design choices. During training, RL achieves the best single-sample performance but reduces multi-sample diversity. At inference time, using the same RL recipe further improves single-shot performance (Sec. 4.1). Repairing incorrect simplifications from execution feedback with Goedel-Prover-V2 effectively corrects errors, but leads to repaired proofs even longer than the originals (Sec. 4.2). Overall, iterative proof shortening offers the best balance between performance and diversity, achieving the strongest results (Sec. 4.3).

Third, we conduct preliminary experiments suggesting two downstream benefits of proof shortening. Training our base model on shortened proofs leads to 2% better performance on miniF2F relative to training on unshortened proofs (Sec. 5.1). Also, shortening proofs often decreases their execution time, with 28% of proofs showing at least a 1.5x speedup after shortening (Sec. 5.2).

**Before ProofOptimizer**        **After ProofOptimizer**

Figure 1: ProofOptimizer reduces the shortest generated proof of a Putnam problem from 1097 to 76 tokens.

## 2    PROOF SIMPLIFICATION: TASK AND METRICS

**Task Definition**   We formalize the proof simplification task as minimizing the complexity of a given proof. Specifically, for a valid formal statement $s$ with proof $p$, the goal is to produce an alternative proof $p^*$ of $s$ that minimizes a complexity measure $\mathcal{L}$:

$$p^* = \arg\min_{x \text{ proves } s} \mathcal{L}(x)$$

Our method is agnostic to the choice of complexity measure $\mathcal{L}$, provided that it is deterministic and can be automatically computed from the proof. This flexibility encompasses the metrics used in prior work (Ahuja et al., 2024). In the rest of this paper, we adopt proof length as the measure of complexity, defined as the number of tokens produced by a Lean-specific tokenizer. Our proof length measure correlates with character count but does not penalize long identifier names, and it ignores comments and line breaks. We denote the length of a proof $x$ by $|x|$, i.e., $\mathcal{L}(x) = |x|$.

**Evaluation Metrics**   Given an original proof $p$ and $k$ candidate simplifications generated by the model, $p'_1, p'_2, \ldots, p'_k$, we define $l_i = \min(|p|, |p'_i|)$ if $p'_i$ is a valid proof and $l_i = |p|$ otherwise. (Intuitively, an invalid attempt reverts to the original proof length). We evaluate proof simplification using two metrics:

- $\min@k \triangleq \min_i \{l_i\}$ denotes the minimum shortened proof length (lower is better).
- $\mathrm{red}@k \triangleq \max_i \left\{ \frac{|p|-l_i}{|p|} \right\} = 1 - \frac{\min@k}{|p|}$ denotes the maximum relative proof length reduction from the original proof (higher is better).

Note that these metrics may not always be correlated: a method that only excels at shortening long proofs has a lower min@k and red@k than one that only excels at shortening short proofs. As with the pass@k metric (Chen et al., 2021), we report our metrics via an unbiased estimator using $n > k$ samples (see Appendix J). We average min@k and red@k across samples in a dataset to get overall length and reduction metrics.

## 3    PROOFOPTIMIZER: LLMS FOR PROOF SIMPLIFICATION

### 3.1   TRAINING

**Lean Base Model**   First, we train a general-purpose Lean model by fine-tuning `Qwen-2.5-7B-Instruct` on a combination of five tasks: natural language problem solving, Lean 4 code completion, auto-formalization (problems and solutions), formal theorem proving, and tactic/proof state prediction.

**Dataset for Proof Simplification**   We employ a four-stage pipeline to generate high-quality proof simplification training data.

1. *Problem Collection*: We first compile a dataset of theorem proving problems from `Goedel-Pset`, filtering out simple computational problems. Each problem consists of a natural language problem, solution, and Lean problem statement.
2. *Proof Sketching*: We train a model that formalizes a problem's natural language solution into a Lean proof sketch consisting of a few high-level proof steps (usually 2-10) with lower level details omitted and filled in with Lean's `sorry` tactic.

3. *Theorem Extraction and Filtering*: For each proof sketch, we extract each proof step into its own separate theorem. At the core, we are taking longer proofs and breaking them down into separate sub-theorems. We collect a total of 518K theorems this way. As we found some of these theorems to be trivial, we design an automation tactic to filter these out, leaving 307K theorems remaining.

4. *Proof Generation*: We use `Goedel-Prover-V2-32B` to generate proofs of these theorems. The model successfully produces Lean proofs of 145K theorems, which we use as our dataset for training.

For more details about our base model and dataset collection, see Appendix B. Next, we describe our two training recipes: expert iteration and online reinforcement learning.

### 3.1.1 PROOFOPTIMIZER-EXPIT: EXPERT ITERATION

We leverage a STaR-like (Zelikman et al., 2022) iterative training algorithm to improve our model. At a high level, we start with our base model $\pi_0$ and the collection of 145K proofs $P_0$. At each iteration, we attempt to simplify each proof, train our model on successful proof simplifications, and use the collection of simplified proofs as seed proofs for the next iteration. More precisely, at each iteration $i$, we do the following:

1. **Sample**: For each proof $x \in P_i$, use $\pi_i$ to sample 4 simplifications $Y_p \triangleq \{y_x^1, y_x^2, y_x^3, y_x^4\} \sim \pi_i(x)$.

2. **Filter**: Use the Lean compiler to find the shortest correct simplification $y_x \in \{x\} \cup Y_x$. Create a training dataset of proof simplifications $D_i = \{(x, y_x) \mid \text{len}(y_x) \leq 0.8 \cdot \text{len}(x), x \in P_i\}$. The length constraint is designed to encourage the model to learn more substantial simplifications rather than trivial ones. For iterations after the first, as $x$ may have been simplified from a more complex proof $x' \in P_0$, we also add $(x', y_x)$ pairs to $D_i$, which are valid and larger proof simplifications. Also, collect simplified proofs $\pi_{i+1} = \{s_x \mid x \in P_i\}$ for the next iteration.

3. **Train**: Fine-tune $\pi_i$ on $D_i$ to get $\pi_{i+1}$.

### 3.1.2 PROOFOPTIMIZER-RL: ONLINE REINFORCEMENT LEARNING

In addition to expert iteration as described in the previous section, we train a proof optimizer model with online reinforcement learning. Using the same dataset as in expert iteration, the reinforcement learning task consists in producing a valid but shorter proof $y$ for a statement given an initial proof $x$. The reward is defined as the relative shortening $R(x, y) = \frac{|x| - |y|}{|x|}$ if $y$ is valid and $|y| \leq |x|$, and $R(x, y) = 0$ otherwise. We employ an asynchronous variant of the GRPO algorithm (Shao et al., 2024) with advantage $A_i = R_i - \frac{1}{k} \sum_{j \leq k} R_j$ baselined with the average reward of $k = 8$ samples, no advantage normalization by standard deviation (Liu et al., 2025b), no KL regularization, and omitting sequences with zero advantage.

## 3.2 INFERENCE-TIME TECHNIQUES

First, we implement a symbolic linter that removes extraneous tactics via Lean's `linter.unusedTactic` linter, which detects tactics that do not change the proof state and provides messages like `'norm_num' tactic does nothing`. We then compare the following techniques on the linted proofs:

- **Test-Time RL**: We use the setup described in Section 3.1.2 and perform reinforcement learning on our two evaluation sets (jointly). Our test-time RL keeps the input proof fixed, meaning improvements occur solely in the model's parameters.

- **Repair with Execution Feedback**: In this scheme, if ProofOptimizer fails to simplify a proof, we collect the execution feedback and ask `Goedel-Prover-V2-32B` to repair the proof with the error messages. Then, we apply the symbolic linter on the new proofs to further shorten successful repairs.

- **Iterative Proof Shortening**: For a given proof, we sample $k$ candidate shortenings and take the shortest correct one. Then, we sample $k$ shortenings of the new proof, take the shortest correct one – and so on.

## 4  EXPERIMENTS

For all evaluations, we use proofs generated by Goedel-Prover-V2 (Lin et al., 2025a) on two popular datasets in formal math, miniF2F (Zheng et al., 2021) and PutnamBench (Tsoukalas et al., 2024). For miniF2F, we use $n = 194$ proofs (average length 334), and for PutnamBench, we use $n = 75$ proofs (average length 1468). More details and examples of proofs in our evaluation set can be found in Appendix G.

### 4.1  EXPERT ITERATION VS. RL VS. TEST-TIME RL

First, we compare our two training schemes: expert iteration and RL. Starting from our Lean base model, we train *ProofOptimizer-ExpIt* by performing three rounds of expert iteration (Sec. 3.1.1) and *ProofOptimizer-RL* by performing online RL (Sec. 3.1.2) after two rounds of expert iteration. Table 1 shows min@k and red@k scores with respect to linted proofs. We observe steady improvements during each round of expert iteration for both @1 and @32 metrics. **Our final model outperforms Gemini-2.5-Pro**, a strong reasoning model, even when given proof state annotations similar to Chain-of-States in ImProver (Ahuja et al., 2024).

Next, we see that **ProofOptimizer-RL significantly improves single sample (@1) metrics at the expense of diversity collapse**, an issue commonly identified during RL training (Gehring et al., 2024; Walder & Karkhanis, 2025; Yue et al., 2025). In Fig. 2 (a, b), we show the evolution of red@1 during training, observing that miniF2F reduction steadily rises while PutnamBench reduction experiences oscillations. This tension is likely because the distribution of training data is more similar in length to miniF2F than PutnamBench, which has a mean proof length of 4x that of the training set.

Finally, we find that test-time RL leads to even further improvements on min@1 and red@1. This is expected, as the model is able to directly tune its weights to learn from successful simplifications at test-time. However, like ProofOptimizer-RL, we observe an even smaller gap between @1 and @32 metrics. In Fig. 2 (c, d), we observe a much more stable evaluation red@1 curve because the distribution gap between the training and evaluation sets is eliminated.

Table 1: **Min@k and Red@k throughout expert iteration and online RL.** Our RL model has strong @1 results, while our ExpIt model has strong @32 results. RL metrics are Gaussian-smoothed.

| Dataset | Category | Model | Min@1 ↓ | Min@32 ↓ | Red@1 ↑ | Red@32 ↑ |
|---|---|---|---|---|---|---|
| miniF2F | | *Linted* | 302 | | 0.0% | |
| | | Gemini-2.5-Pro | 280 | 207 | 24.3% | 57.2% |
| | | Gemini-2.5-Pro + States | 283 | 207 | 26.4% | 58.7% |
| | | Base (7B) | 283 | 202 | 17.6% | 56.2% |
| | *ExpIt* | Base + It 1 | 266 | 178 | 33.4% | 67.0% |
| | | Base + It 2 | 251 | 166 | 45.1% | 70.6% |
| | | *ProofOptimizer-ExpIt* | 241 | **153** | 49.0% | **72.3%** |
| | *RL* | *ProofOptimizer-RL* | **190** | 152 | **63.6%** | 70.9% |
| | | It 2 + Test-Time RL | **160** | 154 | **72.5%** | **73.4%** |
| Putnam Bench | | *Linted* | 1359 | | 0.0% | |
| | | Gemini-2.5-Pro | 1348 | 1303 | 5.5% | 18.0% |
| | | Gemini-2.5-Pro + States | 1371 | 1319 | 6.1% | 19.2% |
| | | Base (7B) | 1341 | 1222 | 3.9% | 20.5% |
| | *ExpIt* | Base + It 1 | 1341 | 1215 | 5.2% | 22.5% |
| | | Base + It 2 | 1335 | 1186 | 6.9% | 24.7% |
| | | *ProofOptimizer-ExpIt* | 1328 | **1161** | 8.2% | **26.3%** |
| | *RL* | *ProofOptimizer-RL* | **1303** | 1258 | **14.9%** | 21.1% |
| | | It 2 + Test-Time RL | **1260** | 1255 | **23.8%** | 24.2% |

Table 2: Step-by-step success rates, revealing the main bottleneck of long repaired proofs.

| Dataset | Simplification | Repair | Shorter than best (before/after linter) |
|---|---|---|---|
| miniF2F | $\frac{7852}{12416}$ (63.2%) | $\frac{2840}{4564}$ (62.2%) | $\frac{76}{2840} \to \frac{137}{2840}$ (2.7% $\to$ 4.8%) |
| PutnamBench | $\frac{1288}{4800}$ (26.8%) | $\frac{613}{3512}$ (17.4%) | $\frac{5}{613} \to \frac{11}{613}$ (0.8% $\to$ 1.8%) |

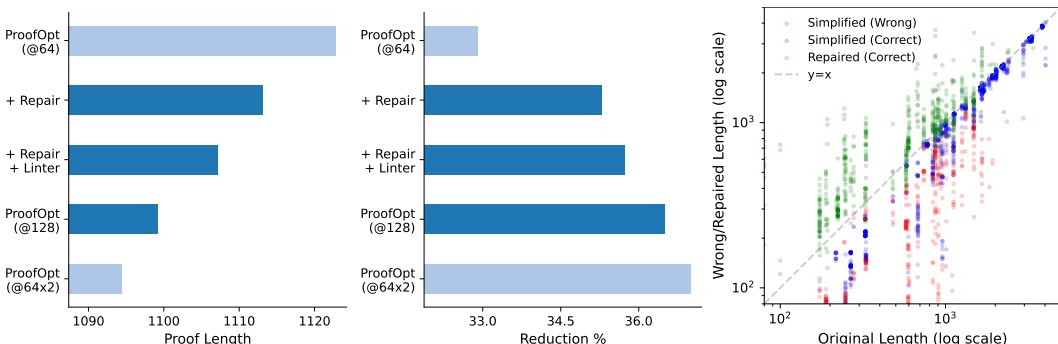

(a) miniF2F (train)   (b) Putnam (train)   (c) miniF2F (test-time)   (d) Putnam (test-time)

Figure 2: **Evolution of proof reduction (red@1) during RL training (a, b) and test-time RL (c, d).** We use Gaussian smoothing ($\sigma = 5$ evaluation intervals for RL training and $\sigma = 3$ for test-time RL). See Fig. 9 for the corresponding red@32 metrics.

## 4.2 ANALYSIS OF REPAIR WITH EXECUTION FEEDBACK

As described in Sec. 3.2, we (1) sample 64 simplifications for each proof with ProofOptimizer-ExpIt, (2) repair incorrect proofs with Goedel-Prover-V2-32B, and (3) shorten successful repairs with our linter. **Overall, we find while repair with execution feedback leads to improvements, it underperforms resampling because repaired proofs are often even longer than the original proofs.** Fig. 3 (left) shows the average proof length and reduction % after sampling, repair, and linting. We our linter to be effective on repaired proofs, decreasing the average repaired proof length from $644 \to 576$ (miniF2F) and $877 \to 788$ (PutnamBench). In Fig. 3 (right), we plot the proof length of the original proofs (before Step 1) against simplified proofs (Step 1) and repaired proofs (Step 2). A majority of the repaired proofs (green dots) are above the $y = x$ line, meaning they are longer than the original proofs, let alone the simplified proofs (blue dots).

Figure 3: Analysis of execution-based repair with Goedel-Prover-V2 on PutnamBench.

In Table 2, we analyze the success rate of each step of our pipeline. However, the key issue remains to be the high length of the repaired proofs. Even after linting, only 4.8% (miniF2F) / 1.8% (Putnam) of post-linted proofs are shorter than the best proof found by ProofOptimizer during simplification. We refer the reader to Appendix F for further analysis and examples.

## 4.3 ITERATIVE PROOF SHORTENING

In Fig. 4 (left), we show the results of iterative proof shortening on miniF2F and PutnamBench proofs using *ProofOptimizer-ExpIt*. First, we do 64 samples per iteration for 6 iterations, observing steady improvement at each iteration. To demonstrate the potential of further scaling, we do 1024 samples

at iterations 7 and 8 and see significant improvement (see Appendix D.2 for analysis on sample size). **Overall, ProofOptimizer combined with iterative proof shortening is very effective on miniF2F and PutnamBench, as average proof length is reduced from** $334 \rightarrow 75$ **and** $1468 \rightarrow 811$**, for an average per-proof reduction of** $87.9\%/57.2\%$**.** In Fig. 4 (right), we plot the overall shortening against the length of the original proof, observing that longer proofs remain challenging to simplify.

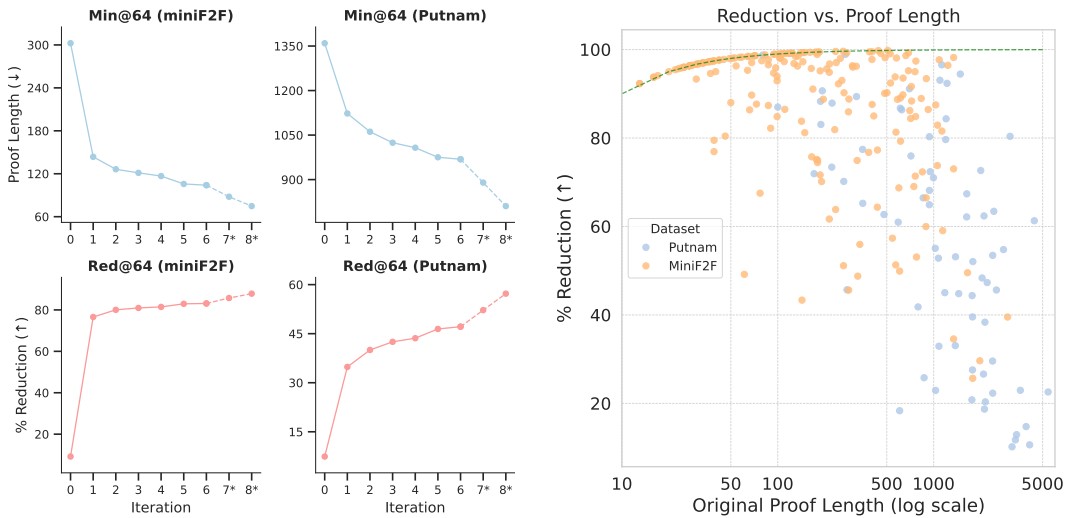

Figure 4: Iterative Shortening: per-iteration improvement (left) and effect of proof length (right)

Finally, in Table 3, we demonstrate the effectiveness of ProofOptimizer on an out-of-distribution dataset, Seed-Prover's four IMO 2025 proofs. With an order of magnitude higher sampling budget, we achieve a significant reduction in the proof length for all four problems, showcasing the potential of our model and technique. Details about our full setup are in Appendix D.3.

Table 3: Iterative shortening achieves significant reduction for Seed-Prover's IMO 2025 proofs.

|  | **P1** | **P3** | **P4** | **P5** |
|---|---|---|---|---|
| Original Proof Length | 36478 | 16377 | 29147 | 8658 |
| Simplified Proof Length | 20506 | 7907 | 14531 | 4002 |
| Length Reduction | 43.8% | 51.7% | 50.1% | 53.8% |

## 5 ADDITIONAL BENEFITS OF PROOF SIMPLIFICATION

### 5.1 TRAINING ON SIMPLIFIED PROOFS IMPROVES GENERATION

Next, we investigate whether fine-tuning on simplified proofs can be advantageous compared to fine-tuning on longer, raw proofs. To do so, we prepare two datasets of identical problems, the first containing a set of proofs generated by `Goedel-Prover-V2` and the second containing the same proofs simplified by ProofOptimizer-ExpIt. The average proof length of the original and simplified proofs is 147 and 85, respectively. We do continued supervised fine-tuning (SFT) starting from our base model (Sec. B.1) with a standard negative log-likelihood (NLL) loss.

In Fig. 5 (left), we compare the training loss between the two datasets. As expected, the initial loss when using original proofs is higher, as models have not seen such long proofs during initial fine-tuning. However, the losses quickly converge. We observe that training on original proofs causes occasional loss spikes, which we suspect are due to several data batches that are hard to learn (e.g. extremely long proofs). Decreasing the learning rate mitigated these training loss spikes but did not improve validation accuracy. In Fig. 5 (right), we compare the miniF2F scores of the two models during SFT, showing that training on simplified proofs results in slightly higher evaluation accuracy despite the two settings having identical training losses.

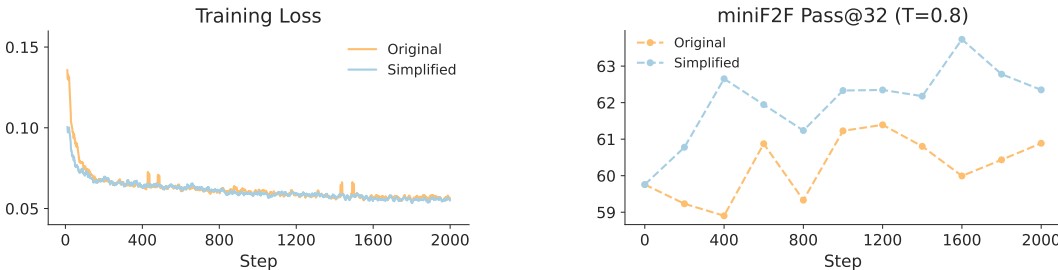

Figure 5: Training loss (left) and miniF2F score (right) after SFT on simplified vs. original proofs.

## 5.2 Simplified Proofs Have a Shorter Execution Time

We also observe that proofs simplified by ProofOptimizer often exhibit a faster execution time. We measure proof execution time with `lake env lean --profile`, excluded library import time (imports are always the same but actual time may vary due to caching effects). We compare the execution times of each proof before and after iterative shortening in Fig. 6 (scatter). For both datasets, we visibly observe that a majority of points lie below the $y = x$ line, signifying speedup. Fig. 6 (histograms) also show the distribution of speedup ratios $\frac{\text{time}_{\text{orig}}}{\text{time}_{\text{new}}}$. Of the 75 PutnamBench proofs, 50/75 have a speedup of over 10%, and 22/75 of those have a speedup of over 50%. We also observe that proofs with a higher original execution time tend to show more speedup. The same trends hold for miniF2F, where 114/194 and 56/194 proofs have a speedup over 10% and 50%, respectively. Finally, we observe 25% and 81% speedups on Seed-Prover's proofs for P3 and P4 of the IMO 2025 (Sec. D.3).

Upon qualitatively analyzing the proofs, we observe that the original proofs often have extraneous tactics that are eliminated by the simplified proofs. However, we also find several cases where the simplified proofs are much slower than the original proof, which usually occurs when a faster proof algorithm is replaced by a shorter but slower method (e.g. brute force with `interval_cases`). We provide two examples of each in Appendix I.2. Finally, we remark that all of our training and inference pipelines can also be applied to proof speedup as well by adjusting the reward function from proof length to proof execution time.

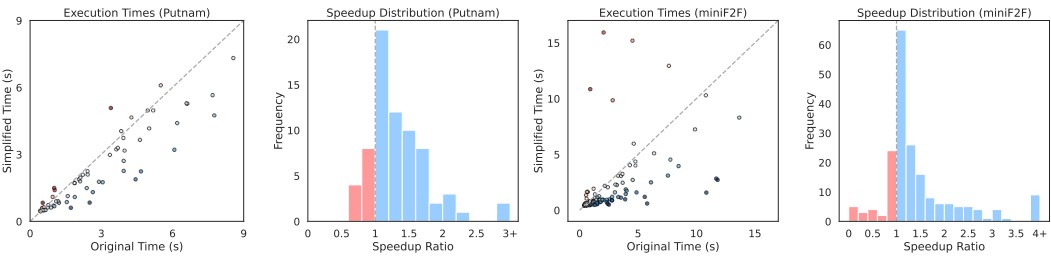

Figure 6: Simplified proofs are frequently faster than original proofs on miniF2F and PutnamBench.

### 5.2.1 Optimizing for Heartbeats instead of Proof Length

As we stated in Sec. 2, our complexity measure $\mathcal{L}$ generalizes beyond proof length. Next, we set $\mathcal{L}$ to be the number of Lean heartbeats[1], a proxy of execution time that can run efficiently in parallel. With this metric, we run eight iterations of the same inference-time algorithm using ProofOptimizer-ExpIt. In Fig. 7 (a, b), we show analogous plots as earlier for miniF2F. Observe that this time, all the points are now on or below the $y = x$ line, eliminating the short but slow proofs we saw in Fig. 6. Overall, we observe faster proofs, with 138/194 and 81/194 miniF2F proofs showing a speedup over 10% and 50%, respectively (compared to 114/194 and 56/194 before using the length metric). In Fig. 7 (c), we see that while the lengths of the proofs found with this metric are slightly longer than before, there is still considerable shortening. Finally, Fig. 7 (d) explains this by showing that proof length and number of heartbeats are generally correlated. In the future, optimizing for a combination of proof

---

[1]We use `#count_heartbeats` with `set_option Elab.async false`

length and heartbeat count could lead to improvements in both readability and execution time. Full results can be found in Sec. I.1.

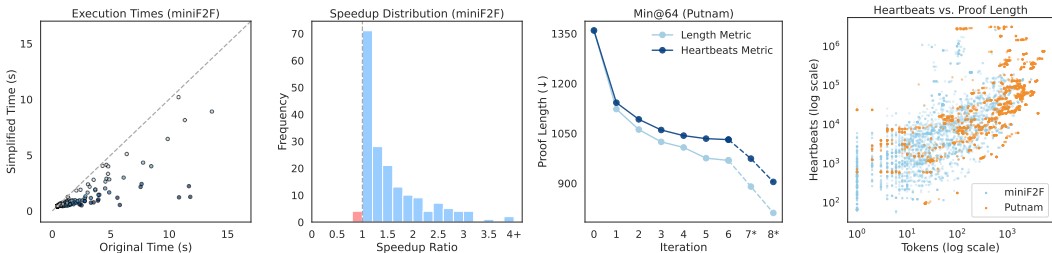

Figure 7: Using heartbeats instead of proof length as complexity measure

## 6    RELATED WORKS

**LLMs for Theorem Proving in Lean**    Formal theorem proving is a rapidly growing frontier in AI for mathematics and software verification (Yang et al., 2024b; Li et al., 2024). Progress is typically measured with benchmarks of mathematical theorems in Lean such as miniF2F (Zheng et al., 2021), PutnamBench (Tsoukalas et al., 2024), and ProofNet (Azerbayev et al., 2023). Recently, there have been many LLMs developed for Lean such as Seed-Prover (Chen et al., 2025), Goedel-Prover (Lin et al., 2025a), DeepSeek-Prover (Ren et al., 2025), and Kimina-Prover (Wang et al., 2025). There have also been post-training techniques built on top of these models, such as with expert iteration (Lin et al., 2024), proof sketching (Cao et al., 2025), tree search (Lample et al., 2022; Zimmer et al., 2025), self-play (Dong & Ma, 2025), proof repair (Ospanov et al., 2025), and RL (Gloeckle et al., 2024).

**AI for Program Simplification**    A related line of work makes programs shorter or more efficient (Schkufza et al., 2013; Mankowitz et al., 2023; Shypula et al., 2023; Gautam et al., 2024). In parallel, library learning aims to discover reusable abstractions, often eliminated repeated code and shortening programs (Ellis et al., 2023; Grand et al., 2023; Kaliszyk & Urban, 2015; Wang et al., 2023; Zhou et al., 2024; Berlot-Attwell et al., 2024). Finally, symbolic reasoning techniques like program slicing (Weiser, 2009), super-optimization (Sasnauskas et al., 2017), or partial evaluation (Jones, 1996) can also shorten and optimize low-level code.

**Automated Proof Shortening**    Frieder et al. (2024) study factors that make Lean proofs easier to understand, motivating shorter proofs for maintainability. Classically, there have also been many symbolic methods targeting shortening proofs in SAT and first-order logic languages  (Rahul & Necula, 2001; Vyskočil et al., 2010; Wernhard & Bibel, 2024; Gladshtein et al., 2024; Kinyon, 2018). On the neural side, GPT-f (Polu & Sutskever, 2020) generated 23 verified proofs shorter than those in the Metamath library. Most related to our work, ImProver (Ahuja et al., 2024), is an inference-time method for proof shortening using GPT-4o with proof states and retrieval. In contrast, we use training-time approaches (expert iteration and RL), analyze complementary inference-time techniques, and focus on shortening longer proofs generated by SoTA LLMs.

## 7    CONCLUSION

We present ProofOptimizer, the first language model trained to simplify Lean proofs. Unlike prior work that wraps existing LLMs around agentic scaffolding, we train a model using expert iteration and RL, coupled with a symbolic linter and iterative proof shortening at inference time. Although simple, our approach already yields nontrivial results, reducing proof length by an average of 87% on MiniF2F, 57% on PutnamBench, and over 50% on Seed-Prover's IMO 2025 proofs. In addition, our methodology, framework, and insights generalize beyond Lean proof shortening and apply to other domains and metrics as well. As AI becomes more tightly integrated with mathematics, we envision a future where AI-generated proofs are not only correct but also concise and readable, with simplification serving as a critical bridge between rigorous formal proofs and human intuitive understanding.

ACKNOWLEDGMENTS

We thank Heather Macbeth for suggesting the experiments in Sec. 5.2.1 and writing the code to count heartbeats, Albert Jiang and Melanie Matchett Wood for discussions regarding desiderata in proof simplification, Amaury Hayat for providing guidance throughout the project, Kunhao Zheng for providing insights on the RL experiments, and many other members at FAIR for various technical contributions, suggestions, and insightful discussions.

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

## A  DISCLOSURE OF USE OF LLMS (ICLR 2026 REQUIREMENT)

In line with the LLM usage disclosure policy for ICLR 2026 submissions, we report our usage of LLMs as the following:

- Design and polish matplotlib and seaborn figures in the paper (ChatGPT)
- Write LaTeX code for tables, figures, and listings, including aesthetically enhancing the styles (ChatGPT)
- Polish and edit text in the paper (ChatGPT)
- Find relevant citations for related work (ChatGPT)
- Assist in producing code for experiments (GitHub Copilot in VSCode, ChatGPT)

## B  LEAN BASE MODEL AND PROOF SIMPLIFICATION DATA DETAILS

### B.1  GENERAL BASE MODEL FOR LEAN

First, we train a general-purpose base model in Lean by fine-tuning `Qwen-2.5-7B-Instruct` (Yang et al., 2024a) on around 1B Lean tokens. The model is fine-tuned on a combination of diverse math and Lean-related tasks, as follows:

- **Natural Language Problem Solving**: The model is trained on natural language mathematics problems with associated solutions so that it has general math capabilities. We use `NuminaMath-1.5` (LI et al., 2024), a high-quality set of such pairs.

- **Lean Code Completion**: We use a subset of Lean code from GitHub, using GPT-4o with heuristics to classify whether code is Lean 3 or Lean 4. We include only the Lean 4 subset of the code.

- **Auto-formalization**: In order to teach the model to associate natural language with Lean, we train the model to perform auto-formalization of both problems and solutions from natural language to Lean 4 in our data mix. For problems, we use natural language problems with Lean problem statement formalizations from high-quality datasets: CombiBench (Liu et al., 2025a), Compfiles, FormalMATH (Yu et al., 2025), Goedel-Pset (Lin et al., 2025a), Lean Workbook (Ying et al., 2024), miniF2F (Zheng et al., 2021), ProofNet (Azerbayev et al., 2023), and PutnamBench (Tsoukalas et al., 2024). We include solution autoformalization data from the `Goedel-Pset-v1-Solved` dataset by mapping Lean solutions with natural language solutions.

- **Formal Theorem Proving**: We use a set of conjectures and proofs from STP (Dong & Ma, 2025), which is a diverse collection of theorems and proofs in Lean 4 generated via expert iteration while training their model.

- **Tactic and Proof State Prediction**: Finally, to teach the model about proof states, we use pre-extracted data from `LeanUniverse` (Aram H. Markosyan, 2024) and extract additional data using the Pantograph (Aniva et al., 2025) tool. For each proof in STP, we extract each tactic, as well as the proof states before and after the tactic. The model is given the proof state before the tactic and asked to predict both the tactic and the proof state following the tactic.

### B.2  GENERATING A DATASET OF THEOREMS AND PROOFS FOR SHORTENING

After creating a Lean base model, we next describe how we generate a training dataset of proofs to be shortened. To do so, we first present a recipe for generating interesting theorems.

**Formalizing Proofs with Sketches to Derive Subtheorems**  While there are many datasets such as `Goedel-Pset` and `Lean Workbook`, we find that they have a high density of simple computational problems posed as proofs rather than high-quality proving problems. In `Goedel-Pset`, we estimate that only 5% of the problems are proof problems[2], leading to a lack of high-quality theorem proving data. To combat this, we develop a technique to generate diverse and interesting theorems based on the idea of proof sketching (Jiang et al., 2022).

The key idea is that we can leverage existing natural language solutions to identify core steps in a proof. We first train our Lean base model to take a natural language solution and auto-formalizing into a high-level proof, which we call a *proof sketch*, an example shown in Listing 1. In the proof sketch, core steps are represented via `have` statements, and lower-level details are omitted and left as `sorry` statements. We then filter sketches are then filtered by the Lean compiler to remove non-compiling sketches.

Once we have a set of compiling sketches, we extract each `sorry` goal into a new theorem via the `extract_goal` tactic, which turns it into a theorem that is equivalent to what needs to be proved at that particular `sorry`. For example, extracting the second `sorry` in Listing 1 results in the theorem shown in Listing 2. By extracting these `sorry` statements, we are able to generate 518K theorems.

---

[2]We estimate whether a problem is a computational problem via a heuristic filter of whether the problem has any of the keywords: *prove, show, establish, demonstrate, verify*

```
theorem lean_workbook_plus_22532  (a b : ℕ → ℝ)
  (h₀ : 0 < a ∧ 0 < b)
  (h₁ : ∀ n, a (n + 1) = a n + 2)
  (h₂ : ∀ n, b (n + 1) = b n * 2)
  (h₃ : a 1 = 1)
  (h₄ : b 1 = 1)
  (h₅ : Σ k in Finset.range 3, b k = 7) :
  Σ k in Finset.range n, (a k * b k) = (2 * n - 3) * 2^n + 3   := by
  -- Lemma 1: Prove that the sequence {a_n} is an arithmetic sequence.
  have lemma1 : ∀ n, a (n + 1) = a n + 2 := by
    sorry

  -- Lemma 2: Express a_n in terms of n.
  have lemma2 : ∀ n, a n = 2 * n - 1 := by
    sorry

  -- Lemma 3: Express b_n in terms of n.
  have lemma3 : ∀ n, b n = 2^(n - 1) := by
    sorry

  -- Lemma 4: Calculate the sum of the first n terms of the sequence {a_n b_n}.
  have lemma4 : ∀ n, Σ k in Finset.range n, (a k * b k) = (2 * n - 3) * 2^n + 3 := by
    sorry

  -- Apply lemma4 to conclude the theorem.
  exact lemma4 n
```

Listing 1: Example of a proof sketch

```
theorem lean_workbook_plus_22532.extracted_1_1 (a b : ℕ → ℝ) (h₀ : 0 < a ∧ 0 < b) (h₁
    ↪ : ∀ (n : ℕ), a (n + 1) = a n + 2)
  (h₂ : ∀ (n : ℕ), b (n + 1) = b n * 2) (h₃ : a 1 = 1) (h₄ : b 1 = 1) (h₅ : Σ k ∈
    ↪ Finset.range 3, b k = 7)
  (lemma1 : ∀ (n : ℕ), a (n + 1) = a n + 2) (n : ℕ) : a n = 2 * ↑n - 1 := sorry
```

Listing 2: Example of an extracted theorem

**Fine-Tuning our Model for Proof Sketching** In order to fine-tune our model for proof sketching, we first curate a dataset of natural language problems (with corresponding Lean problem formalizations) and solutions by combining `Goedel-Pset-v1` (Lin et al., 2025a) with NuminaMath-1.5 (LI et al., 2024). Then, we use `Qwen-2.5-32B-Instruct` to produce proof-sketches based on these natural language solutions similar to that in Listing 1. We filter out compiling sketches and train our Lean base model on them. In Table 4, we show the results of fine-tuning. Since it can be tricky to measure the objective correctness of a sketch, we use the proxy of compile rate, finding our model performs better than `Qwen2.5-32B` and is smaller and can do inference faster.

Table 4: Proof sketching ability of models

| Model | compile@1 | compile@16 |
|---|---|---|
| Qwen2.5 7B (zero-shot) | 3.6 | 7.0 |
| Qwen2.5 7B (one-shot) | 4.9 | 19.0 |
| Qwen2.5 32B (zero-shot) | 21.1 | 62.0 |
| Qwen2.5 32B (one-shot) | 35.1 | 75.0 |
| Ours (7B) | 54.8 | 89.1 |

**Generating Proofs for Simplification** Because proof sketching can generate steps or sub-theorems that are too incremental, we first filter out trivial theorems that can be easily solved by automation tactics in Lean. For example, the first `sorry` in Listing 1 is just a restatement of hypothesis $h_1$ and can be solved via `rfl`. While this theorem is correct, it is not challenging for the model. Therefore, we design an `AUTO` tactic (Listing 3) that tries a series of Lean automation tactics such as `linarith` and `aesop` to filter out these simple theorems, leaving 307K of the original 518K theorems (filtering out 41%).

For the remaining theorems, we attempt to generate proofs of these theorems with `Goedel-Prover-V2-32B`, a strong open-source proving model. With 4 attempts per theorem, the model is able to prove 145K theorems, which we use as targets for proof simplification. Statistics and zerexamples of these proofs can be found in the next section, Appendix B.3.

```
macro "AUTO" : tactic =>
  '(tactic|
    repeat'
      (try rfl
       try tauto
       try assumption
       try norm_num
       try ring
       try ring_nf at *
       try ring_nf! at *
       try native_decide
       try omega
       try simp [*] at *
       try field_simp at *
       try positivity
       try linarith
       try nlinarith
       try exact?
       try aesop))
```

Listing 3: AUTO tactic for filtering trivial theorems

## B.3 STATISTICS OF PROOF SIMPLIFICATION TRAINING DATASET

The minimum, Q1, median, Q3, and maximum proof lengths of our training dataset are 1, 103, 204, 411, and 10958. The mean is 334. In Fig. 8, we show the distribution of lengths, observing its right-skewed nature. Examples of proofs are shown in Listings 4 and 5. Compared to the proofs in our evaluation sets, we observe that training proofs often have more unused hypotheses, as they are derived from extracting the proof state, which may contain hypotheses that are not used for that particular sub-goal.

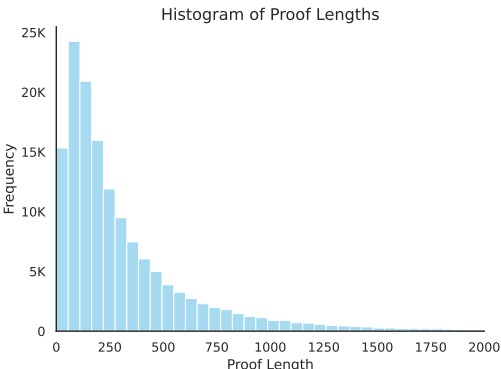

Figure 8: Histogram of proof lengths.

```
theorem extracted_1 (a b : ℝ) (ha : 0 ≤ a) (ha1 : a ≤ 1) (hb : b = a ^ 3 + 1 / (1 + a)
    ↪ )
  (lemma1 : 1 - a + a ^ 2 - a ^ 3 ≤ 1 / (1 + a)) (lemma2 : b ≥ 1 - a + a ^ 2) (lemma3 :
    ↪ 1 - a + a ^ 2 ≥ 3 / 4)
  (lemma4 : b ≤ 3 / 2) : 3 / 4 < b := by
  have h_main : 3 / 4 < b := by
    by_contra h
    -- Assume for contradiction that b ≤ 3/4
    have h₁ : b ≤ 3 / 4 := by linarith
    -- From lemma2, b ≥ 1 - a + ²a, and from lemma3, 1 - a + ²a ≥ 3/4
    have h₂ : 1 - a + a ^ 2 ≤ 3 / 4 := by
      linarith
    -- But from lemma3, 1 - a + ²a ≥ 3/4, so 1 - a + ²a = 3/4
    have h₃ : 1 - a + a ^ 2 = 3 / 4 := by
      linarith
    -- Solve 1 - a + ²a = 3/4 to get a = 1/2
    have h₄ : a = 1 / 2 := by
      have h₄₁ : a ^ 2 - a + 1 / 4 = 0 := by
        nlinarith
      have h₄₂ : (a - 1 / 2) ^ 2 = 0 := by
        nlinarith
      have h₄₃ : a - 1 / 2 = 0 := by
        nlinarith
      linarith
    -- Substitute a = 1/2 into b = ³a + 1/(1 + a)
    have h₅ : b = 19 / 24 := by
      rw [hb]
      rw [h₄]
      norm_num
    -- But 19/24 > 3/4, so b > 3/4, contradiction
    have h₆ : b > 3 / 4 := by
      rw [h₅]
      norm_num
    linarith
  exact h_main
```

Listing 4: Example of Proof Simplification Training Task (Length 158)

```
theorem extracted_1 (n : ℕ) (hn : 3 ≤ n) (lemma1 : Nat.card ↑{k | k ≤ n ∧ k ≠ 0} = n)
    ↪ :
  Nat.card ↑{k | k ≤ n - 1 ∧ k ≠ 0} = n - 1 := by
  have h_main : Nat.card ↑{k : ℕ | k ≤ n - 1 ∧ k ≠ 0} = n - 1 := by
    have h₁ : {k : ℕ | k ≤ n - 1 ∧ k ≠ 0} = Set.Icc 1 (n - 1) := by
      apply Set.ext
      intro k
      simp only [Set.mem_setOf_eq, Set.mem_Icc]
      constructor
      · intro h
        have h₂ : k ≤ n - 1 := h.1
        have h₃ : k ≠ 0 := h.2
        have h₄ : 1 ≤ k := by
          by_contra h₄
          -- If k < 1, then k = 0 since k is a natural number
          have h₅ : k = 0 := by
            omega
          contradiction
        exact ⟨h₄, h₂⟩
      · intro h
        have h₂ : 1 ≤ k := h.1
        have h₃ : k ≤ n - 1 := h.2
        have h₄ : k ≤ n - 1 := h₃
        have h₅ : k ≠ 0 := by
          by_contra h₅
          -- If k = 0, then 1 ≤ k would be false
          have h₆ : k = 0 := by simpa using h₅
          omega
        exact ⟨h₄, h₅⟩
    rw [h₁]
    -- Calculate the cardinality of the set {1, ..., n - 1}
    have h₂ : Nat.card (Set.Icc 1 (n - 1) : Set ℕ) = n - 1 := by
      -- Use the fact that the cardinality of the interval [1, n - 1] is n - 1
      have h₃ : n - 1 ≥ 1 := by
        have h₄ : n ≥ 3 := hn
        omega
      -- Use the formula for the cardinality of the interval [a, b]
      rw [Nat.card_eq_fintype_card]
      -- Use the fact that the cardinality of the interval [1, n - 1] is n - 1
      rw [Fintype.card_ofFinset]
      -- Convert the set to a finset and calculate its cardinality
      <;> simp [Finset.Icc_eq_empty, Finset.card_range, Nat.succ_le_iff]
      <;> cases n with
      | zero => contradiction
      | succ n =>
        cases n with
        | zero => contradiction
        | succ n =>
          cases n with
          | zero => contradiction
          | succ n =>
            simp_all [Finset.Icc_eq_empty, Finset.card_range, Nat.succ_le_iff]
            <;> ring_nf at *
            <;> omega
    rw [h₂]
  exact h_main
```

Listing 5: Example of Proof Simplification Training Task (Length 295)

## C  TRAINING METRICS THROUGHOUT RL

In Section 4.1, we observed that expert iteration leads to higher diversity as witnessed by better @32 metrics, while reinforcement learning with standard reinforcement learning algorithms maximizing expected rewards leads to higher @1 metrics. In Figure 9, we show the evolution of proof shortening red@1 alongside red@32. Initial @32 metrics are slowly distilled into @1, but the improvement on @32 metrics is limited.

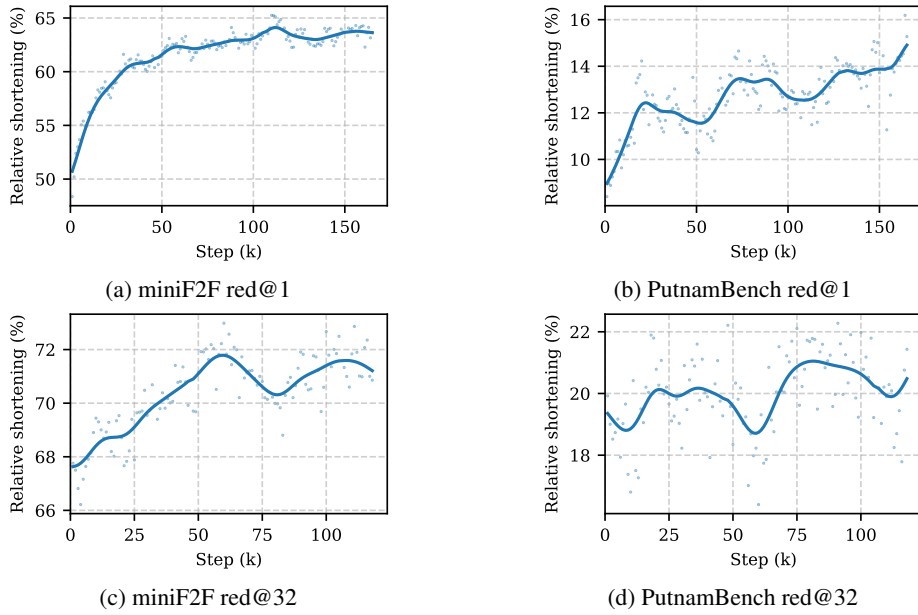

(a) miniF2F red@1

(b) PutnamBench red@1

(c) miniF2F red@32

(d) PutnamBench red@32

Figure 9: **Reduction metrics @1 and @32 over the course of RL**. GRPO maximizes **red@1** at the cost of diversity, as **red@32** only marginally increases in comparison.

# D  FULL RESULTS AND ADDITIONAL ANALYSIS OF ITERATIVE PROOF SHORTENING

## D.1  TABLE OF ITERATIVE PROOF SHORTENING RESULTS

Table 5 is a tabular form of Fig. 4, showing the proof length after each iteration of proof shortening.

Table 5: Min@64 (rounded to nearest integer) and reduction (%) of miniF2F and PutnamBench proofs across inference-time iterations. Iterations $1 - 6$ are done with $64$ samples, and $7 - 8$ with $1024$ samples.

| Dataset | Model | Orig | Lint | It 1 | It 2 | It 3 | It 4 | It 5 | It 6 | It 7* | It 8* |
|---|---|---|---|---|---|---|---|---|---|---|---|
| miniF2F | Min@64 | 334 | 302 | 144 | 126 | 121 | 117 | 106 | 104 | 88 | 75 |
|  | Red@64 (%) | 0.0 | 9.2 | 76.6 | 80.0 | 81.0 | 81.5 | 82.9 | 83.1 | 85.7 | 87.9 |
| Putnam | Min@64 | 1468 | 1359 | 1123 | 1061 | 1024 | 1007 | 975 | 969 | 890 | 811 |
|  | Red@64 (%) | 0.0 | 7.4 | 34.8 | 40.0 | 42.5 | 43.6 | 46.4 | 47.1 | 52.2 | 57.2 |

## D.2  EFFECT OF k ON MIN@K AND RED@K THROUGHOUT SIMPLIFICATION

In this section, we analyze the effect of increasing $k$ on min@k and red@k. First, we analyze this trend when attempting to simplify the initial, linted proof, shown in Table 6 and Fig. 10. We observe a relatively log-linear gain in both metrics.

For comparison, we analyze the same trend but for simplifying proofs that have already gone many iterations of simplification. In Fig. 11, we analyze proofs that have gone 7 iterations of proof simplification. We see a different pattern, where min@k falls slower for lower $k$ and then log-linearly afterwards. Intuitively, as proofs become more simplified, they become harder to simplify in a low-shot setting, and exploring more diverse simplifications becomes crucial.

Table 6: Min@k and Red@k for increasing values of $k$

| Dataset | Metric | Original | Linter | @1 | @2 | @4 | @8 | @16 |
|---|---|---|---|---|---|---|---|---|
| miniF2F | Min@k | 334 | 302 | 142 | 141 | 139 | 137 | 134 |
|  | Red@k (%) | 0.0% | 9.2% | 77.1% | 77.3% | 77.7% | 78.1% | 78.6% |
| PutnamBench | Min@k | 1468 | 1359 | 1120 | 1117 | 1112 | 1105 | 1094 |
|  | Red@k (%) | 0.0% | 7.4% | 35.2% | 35.5% | 35.9% | 36.5% | 37.3% |

| Dataset | Metric | @32 | @64 | @128 | @256 | @512 | @1024 |
|---|---|---|---|---|---|---|---|
| miniF2F | Min@k | 130 | 126 | 122 | 118 | 114 | 110 |
|  | Red@k (%) | 79.2% | 79.9% | 80.6% | 81.2% | 81.8% | 82.4% |
| PutnamBench | Min@k | 1080 | 1063 | 1043 | 1023 | 1004 | 987 |
|  | Red@k (%) | 38.4% | 39.7% | 41.3% | 42.9% | 44.3% | 45.7% |

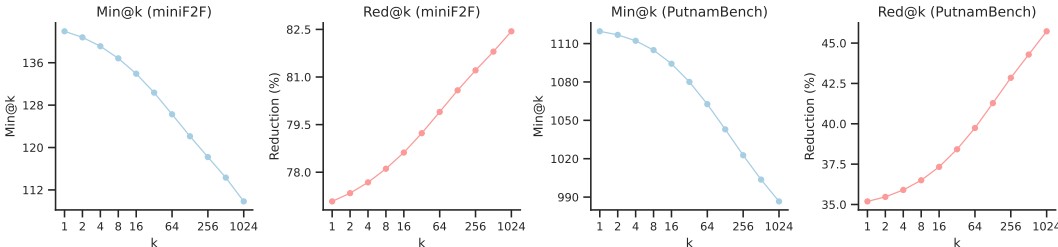

Figure 10: Effect of scaling $k$ (sample count) on Min@k and Red@k (initial iteration)

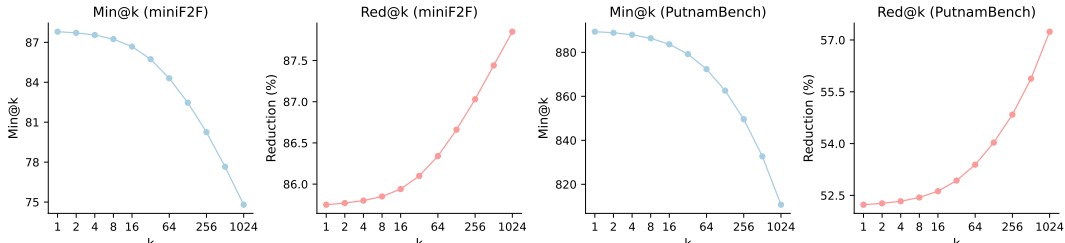

Figure 11: Effect of scaling $k$ (sample count) on Min@k and Red@k (later iteration)

### D.3 DETAILS ON SEED-PROVER IMO PROOF SHORTENING

Earlier in 2025, Seed-Prover released Lean proofs of four problems that the model successfully solved from the 2025 International Mathematical Olympiad (IMO) (Chen et al., 2025). They solved problems 3, 4, and 5 were solved during the contest window, and problem 1 later after the competition. However, the proofs of these problems are extremely verbose, especially compared to their informal counterparts. Using iterative proof shortening, our ProofOptimizer is able to successfully reduce the proof length of their proofs for P3, P4, and P5 by over half, as well as the longer P1 by $43.8\%$. In addition, we find that our shortened proofs for P4 and P5 show a $25\%$ and $81\%$ (respectively) speedup over the original proofs (Table 7).

Table 7: Results for ProofOptimizer + Iterative Shortening on IMO 2025 Proof Simplification

| Problem | Length | | | Runtime | | |
|---------|----------|------------|-----------|----------|------------|----------|
|         | Original | Simplified | Reduction | Original | Simplified | Speedup |
| P1 | 36478 | 20506 | 43.79% | 399.7 | 392.3 | 1.02× |
| P3 | 16377 | 7907  | 51.72% | 39.7  | 39.1  | 1.02× |
| P4 | 29147 | 14531 | 50.15% | 453.8 | 362.5 | 1.25× |
| P5 | 8658  | 4002  | 53.78% | 61.0  | 33.7  | 1.81× |

We use proofs from the official GitHub repository using Mathlib 4.14.0 (our model was trained on Mathlib 4.19.0). Before shortening, we replace invocations of `exact?` and `apply?` with the actual proof that is found. Each of the proofs is divided into a collection of smaller lemmas and theorems (problems 1, 3, 4, and 5 have 80, 52, 88, and 14 theorems, respectively). Since running iterative shortening on the entire proof will suffer from long context issues, we treat each sub-lemma/sub-theorem as an individual target for shortening. At the end, we combine the shortened theorems to produce the complete shortened proof. When feeding a sub-theorem into ProofOptimizer, we include as context the theorem definition (but not proof) of all other theorems that occur in its proof. Finally, to ensure the correctness of our simplified proofs, we use SafeVerify to confirm that all four simplified proofs match the specification of the original proof without any environmental manipulation. We remark that our setup does *not* consider the space of structure-level simplifications, as we retain all sub-theorem statements from the original proof and only simplify their proofs. In addition, as our proof length metric only measures the length of proofs, it does not take into account unnecessarily long or redundant sub-theorem statements.

As this experiment aims to provide a simple demonstration of the potential of our approach rather than perform a controlled scientific study, we do not fix the number of iterations or samples per iteration across problems. Approximately, we use 15-20 iterations of shortening with 64-4096 samples per iteration. Taking inspiration from the analysis in Sec. D.2, we generally use less samples for the first few iterations and increase the number of samples for later iterations to maximize reduction per sample. We also allocate more samples to sub-theorems that show more simplification potential in early iterations. In total, we used approximately 3000 H100 GPU hours per problem.

# E    COMPARISON WITH QWEN2.5, GPT-4O, AND GEMINI-2.5-PRO

In Table 8, we compare *ProofOptimizer* models with several off the shelf models, namely Qwen 2.5 (Team, 2024), GPT-4o (Achiam et al., 2023), and Gemini-2.5-Pro (Comanici et al., 2025). For all models, we feed the output of the symbolic linter as input, and report overall reduction with respect to the *original (unlinted)* proof.

Table 8: **Proof length of miniF2F and PutnamBench proofs for various models.** Specially trained proof minimization models outperform prompted off-the-shelf models. Reinforcement learning achieves best @1 metrics but at the cost of reducing diversity, as witnessed by improved @32 metrics with expert iteration.

| Dataset | Model | Min@1 | Min@32 | Red@1 | Red@32 |
|---------|-------|-------|--------|-------|--------|
| miniF2F | *Original* | 334 | | 0.0% | |
| | *Linter* | 302 | | 9.2% | |
| | Qwen2.5-7B | 294 | 267 | 25.7% | 41.8% |
| | Qwen2.5-32B | 288 | 252 | 30.0% | 47.3% |
| | GPT-4o | 283 | 258 | 35.2% | 47.9% |
| | GPT-4o + States | 266 | 290 | 32.9% | 46.5% |
| | Gemini-2.5-Pro | 280 | 207 | 31.6% | 62.0% |
| | Gemini-2.5-Pro + States | 283 | 208 | 31.6% | 62.0% |
| | ProofOptimizer-ExpIt | 241 | **153** | 53.9% | **74.9%** |
| | ProofOptimizer-RL | **190** | 152 | **67.1%** | 73.4% |
| Putnam Bench | *Original* | 1468 | | 0.0% | |
| | *Linter* | 1359 | | 7.4% | |
| | Qwen2.5-7B | 1358 | 1339 | 9.0% | 14.8% |
| | Qwen2.5-32B | 1353 | 1304 | 10.9% | 20.7% |
| | GPT-4o | 1355 | 1336 | 10.9% | 18.2% |
| | GPT-4o + States | 1379 | 1358 | 9.3% | 15.9% |
| | Gemini-2.5-Pro | 1348 | 1303 | 12.7% | 24.5% |
| | Gemini-2.5-Pro + States | 1371 | 1319 | 11.5% | 24.1% |
| | ProofOptimizer-ExpIt | 1328 | **1161** | 15.2% | **31.9%** |
| | ProofOptimizer-RL | **1303** | 1258 | **21.6%** | 27.1% |

In Fig. 12, we compare the specific optimized proofs between Gemini and ProofOptimizer. For both data sets it can be seen that the longer the proof, the more challenging it is to shorten it. This is because although long proofs have more potential for shortening, the models struggle to maintain correctness of them. Still, ProofOptimizer is able to bring some improvements for the long proofs (see the top right part of the PutnamBench plot). In miniF2F, there is a significant number of proofs that can be minimized to just one step, which typically boils down to invoking one proof automation tactic (like `linarith` instead of applying a sequence of more explicit proof steps.

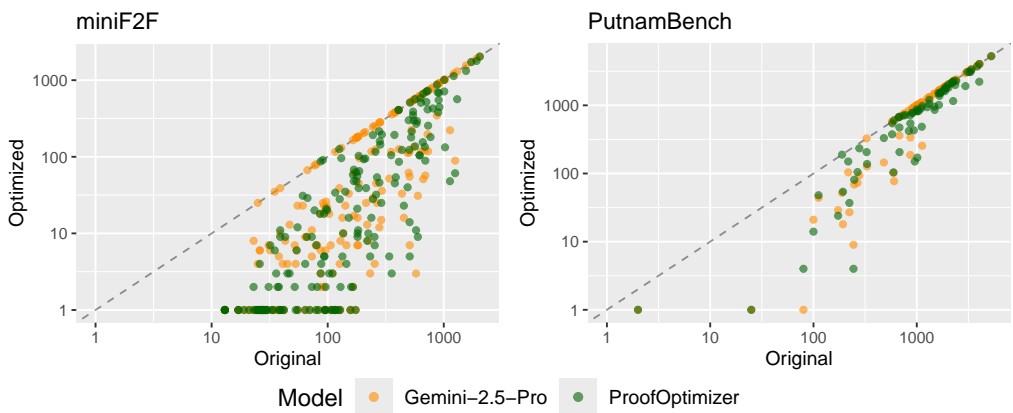

Figure 12: Comparison of optimized proofs between ProofOptimizer (green) and Gemini 2.5 Pro (yellow)

# F    FULL RESULTS AND EXTENDED ANALYSIS OF REPAIR WITH EXECUTION FEEDBACK

This section contains the full results of the experiments in Sec. 4.2. All simplification attempts are done on the set of linted proofs. Table 9, Figure 13, and Figure 14 are extended versions of Fig. 3 for both PutnamBench and miniF2F. The settings are as follows:

- **ProofOptimizer**: *ProofOptimizer-ExpIt*, with 64 simplification attempts per proof.
- **+ Repair**: The previous setting, with 1 attempted repair by `Goedel-Prover-V2-32B`.
- **+ Repair + Linter**: The previous setting, with our linter applied to all proofs.
- **ProofOptimizer (@128)**: *ProofOptimizer-ExpIt*, with 128 simplification attempts per proof
- **ProofOptimizer (@64x2)**: *ProofOptimizer-ExpIt* with 64 simplification attempts per proof, and the best simplified proof for each problem is then fed back for an additional 64 attempts.

We remark that these baselines are normalizing for sample count rather than running time. Sampling a repair from `Goedel-Prover-V2-32B` takes considerably longer than sampling a simplification from our model. This is both because it is a larger model (32B vs. 7B) and because their model relies on CoT, causing their average response length to be significantly longer than ours.

Table 9: Results of execution-based repair strategies

| Dataset | Model | Min@64 | Min@64 × 2 | Red@64 | Red@64 × 2 |
|---------|-------|--------|------------|--------|------------|
| | *Linter* | | *302* | | *9.2%* |
| miniF2F | ProofOptimizer | 144 | - | 75.5% | - |
| | + Repair | - | 136 | - | 77.3% |
| | + Repair + Linter | - | 132 | - | 77.9% |
| | ProofOptimizer (@128) | - | 130 | - | 78.9% |
| | ProofOptimizer (It 2) | - | 125 | - | 80.2% |
| | *Linter* | | *1359* | | *7.4%* |
| Putnam Bench | ProofOptimizer | 1123 | - | 32.9% | - |
| | + Repair | - | 1113 | - | 35.3% |
| | + Repair + Linter | - | 1107.2 | - | 35.7% |
| | ProofOptimizer (@128) | - | 1099 | - | 36.5% |
| | ProofOptimizer (@64x2) | - | 1095 | - | 37.0% |

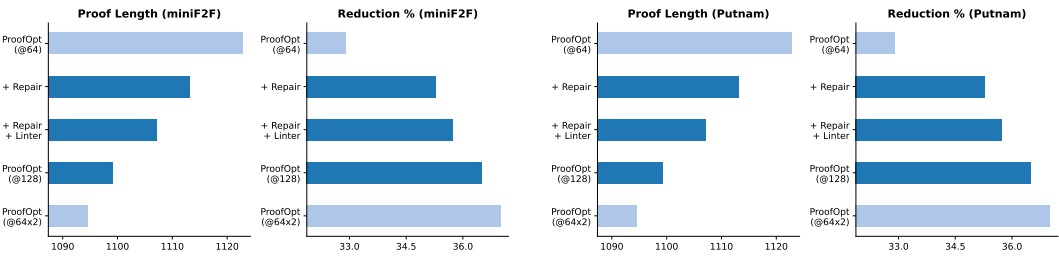

Figure 13: Results of Execution-Based Repair with Goedel-Prover

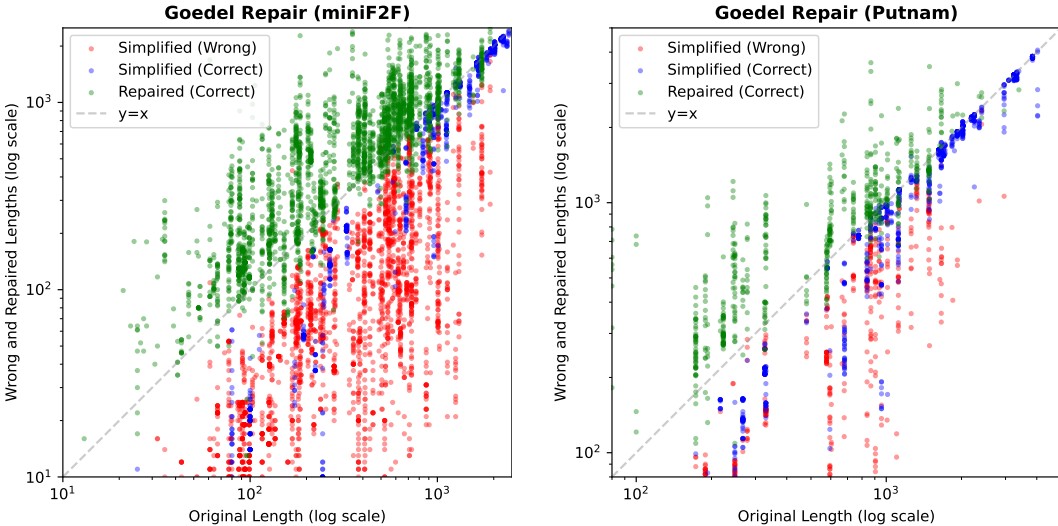

Figure 14: Comparison of Proof Lengths with Execution-Based Repair

Finally, in Listings 6 and 7, we show an example of a proof that was successfully repaired. Note that the repaired proof has many components in common with the original proof.

```
theorem mathd_numbertheory_314
  (r n : ℕ) (h₀ : r = 1342 % 13) (h₁ : 0 <
    ↪ n)
  (h₂ : 1342|n) (h₃ : n % 13 < r)
  : 6710 ≤ n := by
  have h_r : r = 3 := by rw [h₀]
  have h_main : 6710 ≤ n := by
    have h₄ : n % 13 < 3 := by
      rw [h_r] at h₃
      exact h₃
    obtain ⟨k, rfl⟩ := h₂
    have h₅ : 0 < k := by
      by_contra h
      push_neg at h
      have h₆ : k = 0 := by omega
      simp [h₆] at h₁
    by_contra! h₆
    have h₇ : k ≤ 4 := by
      by_contra h₈
      have h₉ : k ≥ 5 := by omega
      have h₁₀ : 1342 * k ≥ 1342 * 5 :=
    ↪ by nlinarith
      omega
    interval_cases k <;> norm_num [Nat.
    ↪ mul_mod, Nat.add_mod, Nat.mod_mod]
    ↪  at h₄ ⊢
  exact h_main
```

Listing 6: Original Lean Proof (Length 126)

```
theorem mathd_numbertheory_314 -- Wrong
  (... statement omitted) := by
  rw [h₀] at h₃
  have : n % 13 < 3 := by omega
  obtain ⟨k, rfl⟩ := h₂
  omega

theorem mathd_numbertheory_314 -- Correct
  (... statement omitted) := by
  have h_r : r = 3 := by
    rw [h₀]
    <;> norm_num
    <;> rfl
  have h_main : 6710 ≤ n := by
    have h₄ : n % 13 < 3 := by
      rw [h_r] at h₃
      exact h₃
    obtain ⟨k, rfl⟩ := h₂
    by_contra! h
    have h₅ : k ≤ 4 := by
      omega
    interval_cases k <;> norm_num [Nat.
    ↪ mul_mod, Nat.add_mod, Nat.mod_mod]
    ↪  at h₄ ⊢ <;>
      (try omega) <;> (try contradiction)
  exact h_main
```

Listing 7: Wrong Simplification and Correct Repair (Length 93)

## G  EVALUATION DATASET DETAILS

For our evaluation datasets, we use miniF2F and PutnamBench proofs sampled from `Goedel-LM/Goedel-Prover-V2-32B`. For miniF2F, we sample with temperature 1 and top-p 0.95. For PutnamBench, we use proofs provided by the team. In both cases, we take the shortest passing proof for each problem in Mathlib 4.19.0, resulting in 194 proofs for miniF2F and 75 proofs for PutnamBench. Table 10 and Figure 15 show summary statistics of our dataset. One sample from each dataset is shown in Listings 8 and 9.

As a sidenote, we observe a discrepency in Goedel-Prover-V2-32B's results with Lean versions. Upon testing their model, we measured 90% (pass@64) and 86 (pass@184) on miniF2F and PutnamBench with Mathlib 4.9, but only 80% (pass@64) and 75 (pass@184) with Mathlib 4.19. In this paper, we use Mathlib 4.19 rather than 4.9, as it is more recent and likely more useful to the Lean community.

Table 10: Summary statistics of proof lengths in evaluation dataset

| Dataset | $n$ | Min | Q1 | Median | Q3 | Max | Mean |
|---|---|---|---|---|---|---|---|
| MiniF2F | 194 | 13 | 64 | 167 | 499 | 2980 | 334 |
| PutnamBench | 75 | 2 | 608 | 1179 | 2110 | 5420 | 1468 |

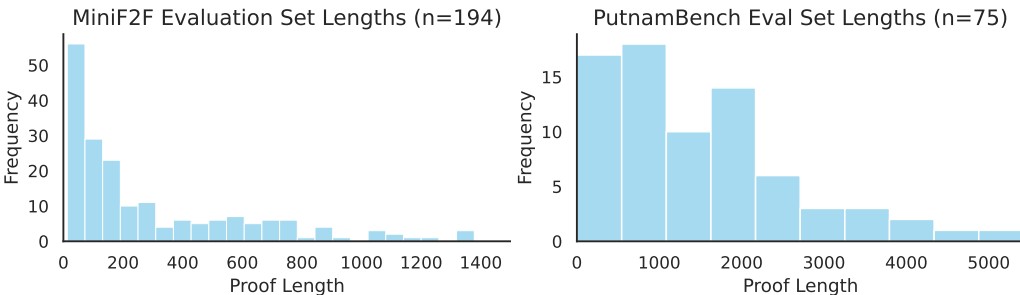

Figure 15: Histograms of proof lengths for our miniF2F and PutnamBench evaluation sets.

```
theorem mathd_numbertheory_185
  (n : ℕ)
  (h₀ : n % 5 = 3) :
  (2 * n) % 5 = 1 := by
  have h₁ : (2 * n) % 5 = 1 := by
    have h₂ : (2 * n) % 5 = (2 * (n % 5)) % 5 := by
      simp [Nat.mul_mod, Nat.mod_mod]
      <;> ring_nf at *
      <;> omega
    rw [h₂]
    rw [h₀]
    <;> norm_num
    <;> rfl

  exact h₁
```

Listing 8: Example of miniF2F Eval Task (Length 65)

```
theorem putnam_1993_a2
(x : ℕ → ℝ)
(xnonzero : ∀ n : ℕ, x n ≠ 0)
(hx : ∀ n ≥ 1, (x n) ^ 2 - x (n - 1) * x (n + 1) = 1)
: ∃ a : ℝ, ∀ n ≥ 1, x (n + 1) = a * x n - x (n - 1) := by
  have h_main : ∀ (n : ℕ), n ≥ 1 → (x (n + 1) + x (n - 1)) / x n = (x 2 + x 0) / x 1
    ↪ := by
    intro n hn
    have h₁ : ∀ (n : ℕ), n ≥ 1 → (x (n + 1) + x (n - 1)) / x n = (x (n + 2) + x n) / x
    ↪  (n + 1) := by
      intro n hn
      have h₂ : (x (n + 1)) ^ 2 - x n * x (n + 2) = 1 := by
        have h₃ := hx (n + 1) (by linarith)
        simpa [Nat.add_assoc] using h₃
      have h₃ : (x n) ^ 2 - x (n - 1) * x (n + 1) = 1 := hx n hn
      have h₄ : x (n + 2) * x n + (x n) ^ 2 - (x (n + 1)) ^ 2 - x (n - 1) * x (n + 1) =
    ↪  0 := by
        linarith
      have h₅ : (x (n + 2) + x n) * x n - (x (n + 1) + x (n - 1)) * x (n + 1) = 0 := by
        ring_nf at h₄ ⊢
        linarith
      have h₆ : x n ≠ 0 := xnonzero n
      have h₇ : x (n + 1) ≠ 0 := xnonzero (n + 1)
      have h₈ : (x (n + 2) + x n) / x (n + 1) - (x (n + 1) + x (n - 1)) / x n = 0 := by
        field_simp [h₆, h₇] at h₅ ⊢
        nlinarith
      linarith

    have h₂ : ∀ (n : ℕ), n ≥ 1 → (x (n + 1) + x (n - 1)) / x n = (x 2 + x 0) / x 1 :=
    ↪ by
      intro n hn
      induction' hn with n hn IH
      ·
        norm_num
      ·
        have h₃ := h₁ n hn
        have h₄ := h₁ (n + 1) (by linarith)
        simp [Nat.add_assoc] at h₃ h₄ ⊢
        <;>
        (try norm_num at * <;>
        try linarith) <;>
        (try simp_all [Nat.add_assoc]) <;>
        (try ring_nf at * <;>
        try linarith) <;>
        (try field_simp [xnonzero] at * <;>
        try nlinarith)
        <;>
        linarith
    exact h₂ n hn

  have h_exists_a : ∃ (a : ℝ), ∀ (n : ℕ), n ≥ 1 → x (n + 1) = a * x n - x (n - 1) :=
    ↪ by
    use (x 2 + x 0) / x 1
    intro n hn
    have h₁ : (x (n + 1) + x (n - 1)) / x n = (x 2 + x 0) / x 1 := h_main n hn
    have h₂ : x n ≠ 0 := xnonzero n
    have h₃ : (x (n + 1) + x (n - 1)) / x n = (x 2 + x 0) / x 1 := by rw [h₁]
    have h₄ : x (n + 1) + x (n - 1) = ((x 2 + x 0) / x 1) * x n := by
      field_simp [h₂] at h₃ ⊢
      <;> nlinarith
    have h₅ : x (n + 1) = ((x 2 + x 0) / x 1) * x n - x (n - 1) := by linarith
    exact h₅

  exact h_exists_a
```

Listing 9: Example of PutnamBench Eval Task (Length 715)

## H    EXAMPLES OF PROOFS SIMPLIFIED BY PROOFOPTIMIZER

In Listings 10 to 17, we show proofs successfully optimized with ProofOptimizer and iterative shortening. Some proofs were syntactically modified to fit on the page (new lines removed, multiple lines compressed into one).

```
theorem mathd_algebra_338 -- Original
    ↪ Proof
  (a b c : ℝ)
  (h₀ : 3 * a + b + c = -3)
  (h₁ : a + 3 * b + c = 9)
  (h₂ : a + b + 3 * c = 19) :
  a * b * c = -56 := by
  have h₃ : b = a + 6 := by
    have h₃₁ : -a + b = 6 := by
      have h₃₂ : (a + 3 * b + c) - (3 * a
  ↪ + b + c) = 9 - (-3) := by
        linarith
      linarith
    linarith

  have h₄ : c = a + 11 := by
    have h₄₁ : -a + c = 11 := by
      have h₄₂ : (a + b + 3 * c) - (3 * a
  ↪ + b + c) = 19 - (-3) := by
        linarith
      linarith
    linarith

  have h₅ : a = -4 := by
    have h₅₁ : 3 * a + b + c = -3 := h₀
    rw [h₃, h₄] at h₅₁
    ring_nf at h₅₁ ⊢
    linarith

  have h₆ : b = 2 := by
    rw [h₃]
    rw [h₅]
    <;> norm_num

  have h₇ : c = 7 := by
    rw [h₄]
    rw [h₅]
    <;> norm_num

  have h₈ : a * b * c = -56 := by
    rw [h₅, h₆, h₇]
    <;> norm_num

  exact h₈
```

Listing 10: Original Proof (Length 214)

```
theorem mathd_algebra_338
  (a b c : ℝ)
  (h₀ : 3 * a + b + c = -3)
  (h₁ : a + 3 * b + c = 9)
  (h₂ : a + b + 3 * c = 19) :
  a * b * c = -56 := by
  have : a = -4 := by linarith
  subst_vars
  nlinarith
```

Listing 11: Simplified Proof (Length 11)

```
theorem putnam_2015_a2
(a : ℕ → ℤ)
(abase : a 0 = 1 ∧ a 1 = 2)
(arec : ∀ n ≥ 2, a n = 4 * a (n - 1) - a (n - 2))
: Odd ((181) : ℕ) ∧ ((181) : ℕ).Prime ∧ ((((181) : ℕ) : ℤ) | a 2015) := by
  constructor
  · decide
  constructor
  · norm_num [Nat.Prime]
  have h₁ : ∀ n : ℕ, (a (n + 10) : ℤ) ≡ - (a n : ℤ) [ZMOD 181] := by
    intro n
    induction' n using Nat.strong_induction_on with n ih
    rcases n with (_ | _ | _ | _ | _ | _ | _ | _ | _ | n) <;>
      simp_all [Int.ModEq, abase, arec] <;> omega
  have h₂ : (a 5 : ℤ) ≡ 0 [ZMOD 181] := by norm_num [Int.ModEq, abase, arec]
  have h₃ : (a 2015 : ℤ) ≡ 0 [ZMOD 181] := by
    have h₄ : ∀ k : ℕ, (a (10 * k + 5) : ℤ) ≡ 0 [ZMOD 181] := by
      intro k
      induction' k with k ih
      · norm_num [Int.ModEq] at h₂ ⊢
        <;> simpa [abase, arec] using h₂
      · have h₅ := h₁ (10 * k + 5)
        have h₆ := h₁ (10 * k + 6)
        have h₇ := h₁ (10 * k + 7)
        have h₈ := h₁ (10 * k + 8)
        have h₉ := h₁ (10 * k + 9)
        have h₁₀ := h₁ (10 * k + 10)
        norm_num [Int.ModEq] at h₅ h₆ h₇ h₈ h₉ h₁₀ ih ⊢
        <;> ring_nf at * <;> omega
    have h₅ : (a 2015 : ℤ) ≡ 0 [ZMOD 181] := by
      have h₆ : (a (10 * 201 + 5) : ℤ) ≡ 0 [ZMOD 181] := h₄ 201
      norm_num at h₆ ⊢
      <;> simpa [add_assoc] using h₆
    exact h₅
  exact Int.dvd_of_emod_eq_zero h₃
```

Listing 12: Original Proof (Length 324)

```
theorem putnam_2015_a2
(a : ℕ → ℤ)
(abase : a 0 = 1 ∧ a 1 = 2)
(arec : ∀ n ≥ 2, a n = 4 * a (n - 1) - a (n - 2))
: Odd ((181) : ℕ) ∧ ((181) : ℕ).Prime ∧ ((((181) : ℕ) : ℤ) | a 2015) := by
  constructor
  · decide
  constructor
  · norm_num [Nat.Prime]
  rw [show 2015 = 10 * 202 - 5 by norm_num]
  have h₁ : ∀ n : ℕ, a (10 * n + 5) ≡ 0 [ZMOD 181] := by
    intro n
    induction' n with k ih
    · norm_num [abase, arec, Int.ModEq]
    · rw [Nat.mul_succ]
      simp_all [Int.ModEq, arec]
      omega
  have h₂ := h₁ 201
  exact Int.dvd_of_emod_eq_zero h₂
```

Listing 13: Simplified Proof (Length 82)

```
theorem imo_1960_p2
  (x : ℝ)
  (h₀ : 0 ≤ 1 + 2 * x)
  (h₁ : (1 - Real.sqrt (1 + 2 * x))^2 ≠ 0)
  (h₂ : (4 * x^2) / (1 - Real.sqrt (1 + 2*x))^2 < 2*x + 9)
  (h₃ : x ≠ 0) :
  -(1 / 2) ≤ x ∧ x < 45 / 8 := by
  constructor
  · nlinarith [sq_nonneg (x + 1 / 2)]
  · set s := Real.sqrt (1 + 2 * x) with hs
    have h₅₁ : 0 ≤ 1 + 2 * x := h₀
    have h₅₂ : s ≥ 0 := Real.sqrt_nonneg _
    have h₅₃ : s ^ 2 = 1 + 2 * x := by
      rw [hs]
      rw [Real.sq_sqrt] <;> linarith
    have h₅₄ : (1 - s) ^ 2 ≠ 0 := by simpa [hs] using h₁
    have h₅₅ : s ≠ 1 := by
      intro h
      have h₅₅₁ : (1 - s) ^ 2 = 0 := by
        rw [h]
        norm_num
      contradiction
    have h₅₆ : (s + 1) ^ 2 * (s - 1) ^ 2 = (s ^ 2 - 1) ^ 2 := by
      ring
    have h₅₇ : (s ^ 2 - 1 : ℝ) ^ 2 = 4 * x ^ 2 := by
      rw [h₅₃]
      ring
    have h₅₈ : (4 : ℝ) * x ^ 2 / (s - 1) ^ 2 = (s + 1) ^ 2 := by
      have h₅₈₁ : (s - 1 : ℝ) ^ 2 ≠ 0 := by
        intro h
        have h₅₈₂ : (1 - s : ℝ) ^ 2 = 0 := by
          calc
            (1 - s : ℝ) ^ 2 = (s - 1 : ℝ) ^ 2 := by ring
            _ = 0 := by rw [h]
        contradiction
      field_simp [h₅₈₁] at h₅₇ ⊢
      nlinarith
    have h₅₉ : (4 : ℝ) * x ^ 2 / (1 - s) ^ 2 = (s + 1) ^ 2 := by
      rw [← h₅₈]
      ring
    nlinarith [sq_nonneg (s - 1)]
```

Listing 14: Original Proof (Length 330)

```
theorem imo_1960_p2
  (x : ℝ)
  (h₀ : 0 ≤ 1 + 2 * x)
  (h₁ : (1 - Real.sqrt (1 + 2 * x))^2 ≠ 0)
  (h₂ : (4 * x^2) / (1 - Real.sqrt (1 + 2*x))^2 < 2*x + 9)
  (h₃ : x ≠ 0) :
  -(1 / 2) ≤ x ∧ x < 45 / 8 := by
  constructor
  · nlinarith [sq_nonneg (x + 1 / 2)]
  · have h₅₇ : (4 : ℝ) * x ^ 2 / (1 - Real.sqrt (1 + 2 * x)) ^ 2 = (1 + Real.sqrt (1 +
      ↪ 2 * x)) ^ 2 := by
      have h₅₈ : (1 - Real.sqrt (1 + 2 * x)) ^ 2 ≠ 0 := by assumption
      field_simp [h₅₈]
      nlinarith [sq_sqrt (show 0 ≤ 1 + 2 * x by assumption)]
    nlinarith [sq_sqrt (show 0 ≤ 1 + 2 * x by assumption),
      Real.sqrt_nonneg (1 + 2 * x)]
```

Listing 15: Simplified Proof (Length 125)

```
theorem putnam_1990_a1
    (T : ℕ → ℤ)
    (hT012 : T 0 = 2 ∧ T 1 = 3 ∧ T 2 = 6)
    (hTn : ∀ n, T (n + 3) = (n + 7) * T (n + 2) − 4 * (n + 3) * T (n + 1) + (4 * n + 4) * T n) :
    T = ((fun n : ℕ => (n)!, fun n : ℕ => 2 ^ n) : (ℕ → ℤ) × (ℕ → ℤ) ).1 + ((fun n : ℕ => (n)!, fun
    ↪ n : ℕ => 2 ^ n) : (ℕ → ℤ) × (ℕ → ℤ) ).2 :=
  by
  have h_main : ∀ (n : ℕ), T n = (n ! : ℤ) + 2 ^ n := by
    intro n
    have h₁ : T n = (n ! : ℤ) + 2 ^ n := by
      have h₂ : ∀ n : ℕ, T n = (n ! : ℤ) + 2 ^ n := by
        intro n
        induction n using Nat.strong_induction_on with
        | h n ih =>
          match n with
          | 0 =>
            norm_num [hT012]
            <;>
            simp_all [Nat.factorial]
            <;>
            norm_num
          | 1 =>
            norm_num [hT012]
            <;>
            simp_all [Nat.factorial]
            <;>
            norm_num
          | 2 =>
            norm_num [hT012]
            <;>
            simp_all [Nat.factorial]
            <;>
            norm_num
          | n + 3 =>
            have h₃ := hTn n
            have h₄ := ih n (by omega)
            have h₅ := ih (n + 1) (by omega)
            have h₆ := ih (n + 2) (by omega)
            simp [h₄, h₅, h₆, pow_add, pow_one, Nat.factorial_succ, Nat.mul_add, Nat.add_mul] at h₃ ⊢
            <;>
            ring_nf at h₃ ⊢ <;>
            norm_cast at h₃ ⊢ <;>
            simp_all [Nat.factorial_succ, pow_add, pow_one, mul_assoc]
            <;>
            ring_nf at * <;>
            norm_num at * <;>
            nlinarith
      exact h₂ n
    exact h₁
  have h_final : T = ((fun n : ℕ => (n)!, fun n : ℕ => 2 ^ n) : (ℕ → ℤ) × (ℕ → ℤ) ).1 + ((fun n : ℕ
    ↪ => (n)!, fun n : ℕ => 2 ^ n) : (ℕ → ℤ) × (ℕ → ℤ) ).2 := by
    funext n
    have h₁ : T n = (n ! : ℤ) + 2 ^ n := h_main n
    simp [h₁, Pi.add_apply]
    <;> norm_cast <;> simp [Nat.cast_add] <;> ring_nf
  apply h_final
```

```
theorem putnam_1990_a1
    (T : ℕ → ℤ)
    (hT012 : T 0 = 2 ∧ T 1 = 3 ∧ T 2 = 6)
    (hTn : ∀ n, T (n + 3) = (n + 7) * T (n + 2) − 4 * (n + 3) * T (n + 1) + (4 * n + 4) * T n) :
    T = ((fun n : ℕ => (n)!, fun n : ℕ => 2 ^ n) : (ℕ → ℤ) × (ℕ → ℤ)).1 + ((fun n : ℕ => (n)!, fun
    ↪ n : ℕ => 2 ^ n) : (ℕ → ℤ) × (ℕ → ℤ)).2 := by
  ext n
  induction' n using Nat.strong_induction_on with n ih
  match n with
  | 0 => simp_all
  | 1 => simp_all
  | 2 => simp_all
  | n + 3 =>
    simp_all [Nat.factorial_succ]
    ring_nf
```

Listing 16: Original Proof (Length 320) and Simplified Proof (Length 34)

```
theorem putnam_1968_a1
: 22/7 - Real.pi = ∫ x in (0)..1, x^4 * (1 - x)^4 / (1 + x^2) := by
  have h_main : (∫ x in (0)..1, x^4 * (1 - x)^4 / (1 + x^2)) = 22/7 - Real.pi := by
    have h_1 : (∫ x in (0)..1, x^4 * (1 - x)^4 / (1 + x^2)) = (∫ x in (0)..1, (x^6 - 4*x^5 + 5*x^4 - 4*
    ↪ x^2 + 4 : ℝ) - 4 / (1 + x^2)) := by
      have h_11 : ∀ (x : ℝ), x^4 * (1 - x)^4 / (1 + x^2) = (x^6 - 4*x^5 + 5*x^4 - 4*x^2 + 4 : ℝ) - 4 /
      ↪ (1 + x^2) := by
        intro x
        have h_12 : (1 + x^2 : ℝ) ≠ 0 := by nlinarith
        have h_13 : x^4 * (1 - x)^4 = (x^6 - 4*x^5 + 5*x^4 - 4*x^2 + 4 : ℝ) * (1 + x^2) - 4 := by
          ring_nf <;> nlinarith [sq_nonneg (x ^ 2), sq_nonneg (x ^ 3), sq_nonneg (x - 1), sq_nonneg (x
        ↪ + 1)]
        have h_14 : x^4 * (1 - x)^4 / (1 + x^2) = ((x^6 - 4*x^5 + 5*x^4 - 4*x^2 + 4 : ℝ) * (1 + x^2) -
        ↪ 4) / (1 + x^2) := by
          rw [h_13]
        rw [h_14]
        field_simp [h_12] <;> ring_nf <;> field_simp [h_12] <;> ring_nf
      congr
      ext x
      rw [h_11 x]
    rw [h_1]
    have h_2 : (∫ x in (0)..1, (x^6 - 4*x^5 + 5*x^4 - 4*x^2 + 4 : ℝ) - 4 / (1 + x^2)) = (∫ x in (0)..1,
    ↪  (x^6 - 4*x^5 + 5*x^4 - 4*x^2 + 4 : ℝ)) - (∫ x in (0)..1, (4 : ℝ) / (1 + x^2)) := by
      apply intervalIntegral.integral_sub
      · apply Continuous.intervalIntegrable
        continuity
      · apply Continuous.intervalIntegrable
        have h_3 : Continuous (fun x : ℝ => (4 : ℝ) / (1 + x ^ 2)) := by
          apply Continuous.div
          · exact continuous_const
          · exact Continuous.add continuous_const (continuous_pow 2)
          · intro x
            have h_4 : (1 + x ^ 2 : ℝ) ≠ 0 := by nlinarith
            exact h_4
        exact h_3
    rw [h_2]
    have h_3 : (∫ x in (0)..1, (x^6 - 4*x^5 + 5*x^4 - 4*x^2 + 4 : ℝ)) = (22 / 7 : ℝ) := by
      norm_num [integral_id, mul_comm] <;> ring_nf <;> norm_num <;> linarith [Real.pi_pos]
    have h_4 : (∫ x in (0)..1, (4 : ℝ) / (1 + x^2)) = Real.pi := by
      have h_41 : (∫ x in (0)..1, (4 : ℝ) / (1 + x ^ 2)) = 4 * (∫ x in (0)..1, (1 : ℝ) / (1 + x ^ 2))
      ↪ := by
        have h_42 : (∫ x in (0)..1, (4 : ℝ) / (1 + x ^ 2)) = (∫ x in (0)..1, 4 * (1 : ℝ) / (1 + x ^ 2))
      ↪  := by
          congr
          ext x <;> ring_nf
        rw [h_42]
        have h_43 : (∫ x in (0)..1, 4 * (1 : ℝ) / (1 + x ^ 2)) = 4 * (∫ x in (0)..1, (1 : ℝ) / (1 + x
      ↪ ^ 2)) := by
          simp [intervalIntegral.integral_comp_mul_left (fun x => (1 : ℝ) / (1 + x ^ 2))] <;>
          norm_num <;> field_simp <;> ring_nf <;> norm_num <;> linarith [Real.pi_pos]
        rw [h_43]
      rw [h_41]
      have h_44 : (∫ x in (0)..1, (1 : ℝ) / (1 + x ^ 2)) = Real.pi / 4 := by
        have h_45 : (∫ x in (0)..1, (1 : ℝ) / (1 + x ^ 2)) = Real.arctan 1 - Real.arctan 0 := by
          rw [integral_one_div_one_add_sq] <;> norm_num
        rw [h_45]
        have h_46 : Real.arctan 1 = Real.pi / 4 := by
          norm_num [Real.arctan_one]
        have h_47 : Real.arctan 0 = 0 := by
          norm_num [Real.arctan_zero]
        rw [h_46, h_47] <;> ring_nf <;> norm_num
      rw [h_44] <;> ring_nf <;> norm_num
    rw [h_3, h_4] <;> ring_nf <;> norm_num
  have h_final : 22/7 - Real.pi = ∫ x in (0)..1, x^4 * (1 - x)^4 / (1 + x^2) := by
    rw [h_main] <;> linarith [Real.pi_pos]
  exact h_final
```

```
theorem putnam_1968_a1
: 22/7 - Real.pi = ∫ x in (0)..1, x^4 * (1 - x)^4 / (1 + x^2) := by
  simp_rw [show ∀ x : ℝ, x ^ 4 * (1 - x) ^ 4 / (1 + x^2) = (x ^6 - 4 * x ^5 + 5 * x ^4 - 4 * x ^2 + 4
  ↪  - 4 / (1 + x ^2)) by
    intro x
    field_simp
    ring]
  ring_nf
  norm_num
  <;> linarith [Real.pi_pos]
```

Listing 17: Original Proof (Length 1097) and Simplified Proof (Length 76)

# I  PROOF SPEEDUP AND SLOWDOWN ANALYSIS AND EXAMPLES

## I.1  ITERATIVE PROOF SHORTENING RESULTS WITH HEARTBEAT METRIC

Table 12 and Fig. 17 show the results of iterative proof shortening using proof length vs. heartbeats as optimization metrics. Observe that while optimizing for heartbeats isn't nearly as effective for proof length, it still leads to considerable simplification.

Table 11: Comparison of Min@64 (rounded to nearest integer), reduction (%), Heartbeats@64 (in thousands), and reduction (%) across inference-time iterations for miniF2F and PutnamBench proofs. Iterations 1–6 use 64 samples, and 7–8 use 1024 samples. The first group shows the standard (length-optimized) setting; the second group shows the new (heartbeat-optimized) experiment.

| Dataset | Metric | Orig | Lint | It 1 | It 2 | It 3 | It 4 | It 5 | It 6 | It 7* | It 8* |
|---|---|---|---|---|---|---|---|---|---|---|---|
| | | | | *Optimizing for Length* | | | | | | | |
| | Min@64 | 334 | 302 | 144 | 126 | 121 | 117 | 106 | 104 | 88 | 75 |
| | Red@64 (%) | 0.0 | 9.2 | 76.6 | 80.0 | 81.0 | 81.5 | 82.9 | 83.1 | 85.7 | 87.9 |
| miniF2F | | | | *Optimizing for Heartbeats* | | | | | | | |
| | Min@64 | 334 | 302 | 163 | 145 | 139 | 135 | 129 | 125 | 112 | 96 |
| | Red@64 (%) | 0.0 | 9.2 | 71.3 | 74.8 | 75.8 | 76.3 | 76.9 | 77.4 | 79.0 | 81.3 |
| | HB@64 (K) | 36.3 | 36.2 | 14.5 | 13.6 | 13.3 | 13.2 | 13.0 | 12.8 | 11.9 | 10.4 |
| | HB Red@64 | 0.0 | 0.2 | 43.3 | 46.7 | 48.2 | 48.5 | 48.8 | 49.6 | 51.5 | 57.0 |
| | | | | *Optimizing for Length* | | | | | | | |
| | Min@64 | 1468 | 1359 | 1123 | 1061 | 1024 | 1007 | 975 | 969 | 890 | 811 |
| | Red@64 (%) | 0.0 | 7.4 | 34.8 | 40.0 | 42.5 | 43.6 | 46.4 | 47.1 | 52.2 | 57.2 |
| Putnam | | | | *Optimizing for Heartbeats* | | | | | | | |
| | Min@64 | 1468 | 1359 | 1142 | 1092 | 1060 | 1043 | 1034 | 1031 | 974 | 904 |
| | Red@64 (%) | 0.0 | 7.4 | 32.2 | 36.2 | 38.7 | 39.7 | 40.5 | 40.8 | 44.0 | 49.2 |
| | HB@64 (K) | 221 | 219 | 199 | 157 | 155 | 140 | 136 | 136 | 122 | 111 |
| | HB Red@64 | 0.0 | 0.7 | 18.5 | 23.9 | 26.9 | 28.4 | 29.5 | 29.6 | 34.0 | 39.5 |

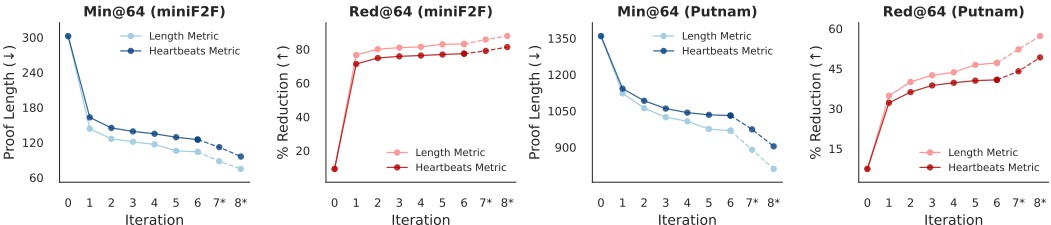

Figure 16: Optimizing for length vs. heartbeats

## I.2  EXAMPLES OF PROOF SPEEDUP AND SLOWDOWN AFTER SIMPLIFICATION

Table 12 and Fig. 17 show the results of iterative proof shortening using proof length vs. heartbeats as optimization metrics. Observe that while optimizing for heartbeats isn't nearly as effective for proof length, it still leads to considerable simplification.

Table 12: Comparison of Min@64 (rounded to nearest integer), reduction (%), Heartbeats@64 (in thousands), and reduction (%) across inference-time iterations for miniF2F and PutnamBench proofs. Iterations 1–6 use 64 samples, and 7–8 use 1024 samples. The first group shows the standard (length-optimized) setting; the second group shows the new (heartbeat-optimized) experiment.

| Dataset | Metric | Orig | Lint | It 1 | It 2 | It 3 | It 4 | It 5 | It 6 | It 7* | It 8* |
|---|---|---|---|---|---|---|---|---|---|---|---|
| | *Optimizing for Length* | | | | | | | | | | |
| | Min@64 | 334 | 302 | 144 | 126 | 121 | 117 | 106 | 104 | 88 | 75 |
| | Red@64 (%) | 0.0 | 9.2 | 76.6 | 80.0 | 81.0 | 81.5 | 82.9 | 83.1 | 85.7 | 87.9 |
| miniF2F | *Optimizing for Heartbeats* | | | | | | | | | | |
| | Min@64 | 334 | 302 | 163 | 145 | 139 | 135 | 129 | 125 | 112 | 96 |
| | Red@64 (%) | 0.0 | 9.2 | 71.3 | 74.8 | 75.8 | 76.3 | 76.9 | 77.4 | 79.0 | 81.3 |
| | HB@64 (K) | 36.3 | 36.2 | 14.5 | 13.6 | 13.3 | 13.2 | 13.0 | 12.8 | 11.9 | 10.4 |
| | HB Red@64 | 0.0 | 0.2 | 43.3 | 46.7 | 48.2 | 48.5 | 48.8 | 49.6 | 51.5 | 57.0 |
| | *Optimizing for Length* | | | | | | | | | | |
| | Min@64 | 1468 | 1359 | 1123 | 1061 | 1024 | 1007 | 975 | 969 | 890 | 811 |
| | Red@64 (%) | 0.0 | 7.4 | 34.8 | 40.0 | 42.5 | 43.6 | 46.4 | 47.1 | 52.2 | 57.2 |
| Putnam | *Optimizing for Heartbeats* | | | | | | | | | | |
| | Min@64 | 1468 | 1359 | 1142 | 1092 | 1060 | 1043 | 1034 | 1031 | 974 | 904 |
| | Red@64 (%) | 0.0 | 7.4 | 32.2 | 36.2 | 38.7 | 39.7 | 40.5 | 40.8 | 44.0 | 49.2 |
| | HB@64 (K) | 221 | 219 | 199 | 157 | 155 | 140 | 136 | 136 | 122 | 111 |
| | HB Red@64 | 0.0 | 0.7 | 18.5 | 23.9 | 26.9 | 28.4 | 29.5 | 29.6 | 34.0 | 39.5 |

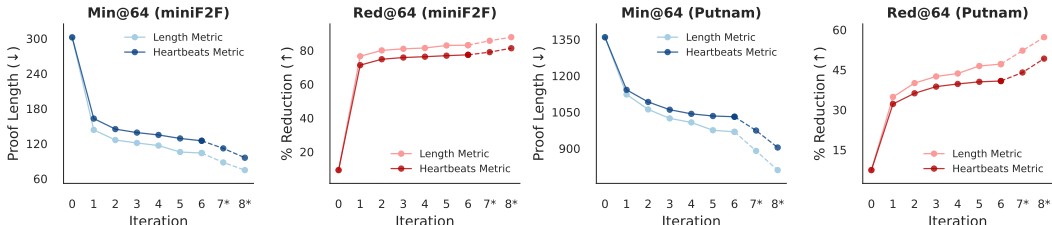

Figure 17: Optimizing for length vs. heartbeats

We analyze two examples of proof speedup and slowdown. In Listing 18, we observe that the original proof uses an extraneous amount of tactics within `nlinarith` in order to prove the main conjecture. By removing a majority of these, the simplified proof achieves a 4.7x speedup. In Listing 19, we observe a more extreme case, where the original proof is significantly overcomplicated and can be reduced to one `omega` invocation. `Goedel-Prover-V2-32B` never found this single-tactic proof (with 64 samples) and instead produces proofs with many unnecessary subgoals, leading to a proof with slow execution time.

In several occurrences, we observe that simplified proofs can be significantly slower than the original proof. This is usually because the simplified proof is notationally shorter, but uses a slower approach to complete the proof. For example, in Listing 20, ProofOptimizer finds a shorter proof, but the proof is reliant on `simp_all`, `Finset.sum_range_succ`, and `linarith`, which expand the goal into large proof terms that are time-consuming, causing the new proof to be over $10\times$ slower. Another example is shown in Listing 21. Here, the original proof first iterates over all $m \leq 71$ with `interval_cases m`, tries to simplify using `omega`, and then iterates over all $n \leq 71$ with `interval_cases n`. ProofOptimizer, however, removes the `try omega`, directly doing an exhaustive search over $(m, n)$. The `try omega` statement in the original proof made it much faster, removing 69 of the 71 goals, whereas the simplified proof had to iterate through $n$ for these goals.

```
theorem imo_1983_p6 -- Original Proof, Time: 5.57s
  (a b c : ℝ)
  (h₀ : 0 < a ∧ 0 < b ∧ 0 < c)
  (h₁ : c < a + b)
  (h₂ : b < a + c)
  (h₃ : a < b + c) :
  0 ≤ a^2 * b * (a - b) + b^2 * c * (b - c) + c^2 * a * (c - a) := by
  have h_main : 0 ≤ a^2 * b * (a - b) + b^2 * c * (b - c) + c^2 * a * (c - a) := by
    nlinarith [sq_nonneg (a - b), sq_nonneg (b - c), sq_nonneg (c - a),
      mul_nonneg h₀.1.le h₀.2.1.le, mul_nonneg h₀.2.1.le h₀.2.2.le, mul_nonneg h₀.2.2.le
      ↪ h₀.1.le,
      mul_nonneg (sq_nonneg (a - b)) h₀.2.2.le, mul_nonneg (sq_nonneg (b - c)) h₀.1.le,
      mul_nonneg (sq_nonneg (c - a)) h₀.2.1.le, mul_pos h₀.1 h₀.2.1, mul_pos h₀.2.1 h₀
      ↪ .2.2,
      mul_pos h₀.2.2 h₀.1, mul_pos (sub_pos.mpr h₁) (sub_pos.mpr h₂),
      mul_pos (sub_pos.mpr h₂) (sub_pos.mpr h₃), mul_pos (sub_pos.mpr h₃) (sub_pos.mpr h
      ↪ ₁),
      sq_nonneg (a + b - 2 * c), sq_nonneg (b + c - 2 * a), sq_nonneg (c + a - 2 * b)]
  exact h_main

theorem imo_1983_p6 -- Simplified Proof, Time: 1.20s
  (a b c : ℝ)
  (h₀ : 0 < a ∧ 0 < b ∧ 0 < c)
  (h₁ : c < a + b)
  (h₂ : b < a + c)
  (h₃ : a < b + c) :
  0 ≤ a^2 * b * (a - b) + b^2 * c * (b - c) + c^2 * a * (c - a) := by
  nlinarith [mul_pos (sub_pos.mpr h₁) (sub_pos.mpr h₂), mul_pos (sub_pos.mpr h₂) (
    ↪ sub_pos.mpr h₃), mul_pos (sub_pos.mpr h₃) (sub_pos.mpr h₁), sq_nonneg (a - b),
    ↪ sq_nonneg (b - c), sq_nonneg (c - a)]
```

Listing 18: Example of Speedup after Simplification (orig: 5.6s, new: 1.2s)

```
theorem mathd_numbertheory_765 -- Original Proof, Time: 2.50s
  (x : ℤ)
  (h₀ : x < 0)
  (h₁ : (24 * x) % 1199 = 15) :
  x ≤ -449 := by
  have h_main : x ≤ -449 := by
    by_contra! h
    have h₂ : -448 ≤ x := by linarith
    have h₃ : x < 0 := h₀
    have h₄ : (24 * x) % 1199 = 15 := h₁
    have h₅ : x ≥ -448 := by linarith
    have h₆ : x ≤ -1 := by
      omega
    -- We will check all possible values of x from -448 to -1 and show that none
    ↪ satisfy (24 * x) % 1199 = 15
    have h₇ : False := by
      -- Use the fact that x is between -448 and -1 to check each possible value
      have h₈ : x ≥ -448 := by linarith
      have h₉ : x ≤ -1 := by omega
      -- Use interval_cases to check each possible value of x
      interval_cases x <;> norm_num [Int.mul_emod, Int.add_emod] at h₄ ⊢ <;> omega
    exact h₇
  exact h_main

theorem mathd_numbertheory_765 -- Simplified Proof, Time: 0.50s
  (x : ℤ)
  (h₀ : x < 0)
  (h₁ : (24 * x) % 1199 = 15) :
  x ≤ -449 := by
  omega
```

Listing 19: Example of Speedup after Simplification (orig: 2.5s, new: 0.5s)

```
theorem aime_1984_p1 -- Original Proof, Time: 0.91s
  (u : ℕ → ℚ)
  (h₀ : ∀ n, u (n + 1) = u n + 1)
  (h₁ : Σ k ∈ Finset.range 98, u k.succ = 137) :
  Σ k ∈ Finset.range 49, u (2 * k.succ) = 93 := by
  have h₂ : ∀ (n : ℕ), u n = u 0 + n := by
    (... 14 lines omitted)

  have h₃ : 98 * u 0 + 4851 = 137 := by
    have h₄ : Σ k in Finset.range 98, u (k.succ) = 137 := h₁
    have h₅ : Σ k in Finset.range 98, u (k.succ) = Σ k in Finset.range 98, (u 0 + (k.
    ↪ succ : ℚ)) := by
      apply Finset.sum_congr rfl
      intro k _
      rw [h₂ (k.succ)]
      <;> simp [Nat.cast_add, Nat.cast_one]
      <;> ring_nf
      <;> norm_num
    rw [h₅] at h₄
    have h₆ : Σ k in Finset.range 98, (u 0 + (k.succ : ℚ)) = 98 * u 0 + 4851 := by
      have h₇ : Σ k in Finset.range 98, (u 0 + (k.succ : ℚ)) = Σ k in Finset.range 98,
      ↪   (u 0 : ℚ) + Σ k in Finset.range 98, (k.succ : ℚ) := by
        rw [Finset.sum_add_distrib]
      rw [h₇]
      have h₈ : Σ k in Finset.range 98, (u 0 : ℚ) = 98 * u 0 := by
        simp [Finset.sum_const, Finset.card_range]
        <;> ring_nf
      rw [h₈]
      have h₉ : Σ k in Finset.range 98, (k.succ : ℚ) = 4851 := by
        norm_num [Finset.sum_range_succ, Finset.sum_range_succ, Finset.sum_range_succ]
        <;>
        rfl
      rw [h₉]
      <;> ring_nf
    rw [h₆] at h₄
    norm_num at h₄ ⊢
    <;> linarith

  have h₄ : Σ k ∈ Finset.range 49, u (2 * k.succ) = 49 * u 0 + 2450 := by
    -- (... 25 lines omitted)
  have h₅ : 49 * u 0 = -2357 := by
    -- (... 6 lines omitted)
  have h₆ : Σ k ∈ Finset.range 49, u (2 * k.succ) = 93 := by
    -- (... 4 lines omitted)
    linarith
  exact h₆

theorem aime_1984_p1 -- Simplified Proof, Time: 10.84s
  (u : ℕ → ℚ)
  (h₀ : ∀ n, u (n + 1) = u n + 1)
  (h₁ : Σ k ∈ Finset.range 98, u k.succ = 137) :
  Σ k ∈ Finset.range 49, u (2 * k.succ) = 93 := by
  simp_all [Finset.sum_range_succ]
  linarith
```

Listing 20: Example of Slowdown after Simplification (orig: 0.9s, new: 10.8s)

```
theorem mathd_numbertheory_711 -- Original Proof, 4.87s
  (m n : ℕ)
  (h₀ : 0 < m ∧ 0 < n)
  (h₁ : Nat.gcd m n = 8)
  (h₂ : Nat.lcm m n = 112) :
  72 ≤ m + n := by
  have h_product : m * n = 896 := by
    -- (... 5 lines omitted)
  have h_main : 72 ≤ m + n := by
    have h₃ : 0 < m := h₀.1
    have h₄ : 0 < n := h₀.2
    have h₅ : m * n = 896 := h_product
    have h₆ : Nat.gcd m n = 8 := h₁
    have h₇ : Nat.lcm m n = 112 := h₂
    have h₈ : m + n ≥ 72 := by
      by_contra! h
      -- (... 4 lines omitted)
      have h₁₁ : m ≤ 71 := by nlinarith
      have h₁₂ : n ≤ 71 := by nlinarith
      interval_cases m <;> norm_num at h₅ ⊢ <;>
        (try omega) <;>
        (try {
          interval_cases n <;> norm_num at h₅ h₆ h₇ ⊢ <;>
          -- (... 5 lines omitted)
        }) <;>
        -- (... 5 lines omitted)
    exact h₈
  exact h_main

theorem mathd_numbertheory_711 -- Simplified Proof, 74.63s
  (m n : ℕ)
  (h₀ : 0 < m ∧ 0 < n)
  (h₁ : Nat.gcd m n = 8)
  (h₂ : Nat.lcm m n = 112) :
  72 ≤ m + n := by
  have : m * n = 896 := by
    rw [← Nat.gcd_mul_lcm m n]
    simp_all
  by_contra!
  have : m ≤ 71 := by nlinarith
  have : n ≤ 71 := by nlinarith
  interval_cases m <;> interval_cases n <;> simp_all
```

Listing 21: Example of Slowdown after Simplification (orig: 4.9s, new: 74.6s)

## J  DERIVATION OF CLOSED FORM FOR MIN@K AND MAX@K

In this section, we derive the closed form expression we use for estimating max@k from $n$ samples based off the classic pass@k metric:

$$\max@k = \frac{1}{\binom{n}{k}} \sum_{i \leq n} \binom{i-1}{k-1} x_i.$$

Let $X$ be a real random variable, $X_1, \ldots, X_k$ independent realizations of $X$ and $X_{(k)} = \max_{i \leq k} X_i$ their maximum. We would like to give an estimator for $\mathbb{E}[X_{(k)}]$ given $n \geq k$ independent samples $x_1 \leq \ldots \leq x_n$ of $X$ sorted by size.

Consider the estimator $M = \frac{1}{\binom{n}{k}} \sum_{i \leq n} \binom{i-1}{k-1} x_i$, with the idea being that there exist $\binom{n}{k}$ ways to choose $k$ out of the $n$ samples overall, out of which $\binom{i-1}{k-1}$ select the $i$-th and then $k-1$ with a smaller index.

We compute

$$
\mathbb{E}_{x_i} \left[ \frac{1}{\binom{n}{k}} \sum_{i \leq n} \binom{i-1}{k-1} x_i \right] = \mathbb{E}_{x_i} \left[ \frac{1}{\binom{n}{k}} \sum_{I \subseteq \{1,\ldots,n\}, |I|=k} x_{\max I} \right]
$$

$$
= \frac{1}{\binom{n}{k}} \sum_{I \subseteq \{1,\ldots,n\}, |I|=k} \mathbb{E}_{x_i} \left[ x_{\max I} \right]
$$

$$
= \frac{1}{\binom{n}{k}} \sum_{I \subseteq \{1,\ldots,n\}, |I|=k} \mathbb{E}_{x_i} \left[ \max_{j \in I} x_j \right]
$$

$$
= \frac{1}{\binom{n}{k}} \sum_{I \subseteq \{1,\ldots,n\}, |I|=k} \mathbb{E} \left[ X_{(k)} \right]
$$

$$
= \mathbb{E} \left[ X_{(k)} \right]
$$

by the counting argument explained above, linearity of expectation, ordering of the $x_i$ and independence.

Note that this is a generalization of the pass@k metric, which covers the case of Bernoulli distributed $X$ (Chen et al., 2021).

We recommend using a numerically stable implementation that computes the ratio $\frac{\binom{i-1}{k-1}}{\binom{n}{k}}$ by canceling a $(k-1)!$ factor and pairing up numerator and denominator factors.

Moreover, the min@k estimator can be obtained as $\min@k(x_1, \ldots, x_n) = -\max@k(-x_1, \ldots, -x_n)$.

## K  HYPERPARAMETERS

In this section, we detail the hyperparameters we use throughout our various training and inference experiments. Prompts can be found in the next section, Appendix L.

**Iterative Training (Sec. 3.1.1)**: For each round of SFT, we use an effective batch size of 64 (2 nodes, 8 H100/node, 4 gradient accumulation steps) and learning rate 1e-5. We use a cosine scheduler with minimum learning rate 1e-8 and 100 steps of warm-up starting from 1e-30. For inference, we use $\tau = 1.0$ and top-p 0.95.

**Reinforcement learning (Sec 3.1.2)**: Our setup is asynchronous online reinforcement learning with 16 trainer and 16 worker GPUs, and 16 environment copies per worker GPU. We use a global training batch size of 32 (local batch size 2 per trainer), a constant learning rate of 6e-8 following a linear warmup over 200 steps, a GRPO group size of 8, mean normalization but no variance normalziation, no KL penalty and model updates sent to workers every 100 steps. Workers use For inference, we use $\tau = 1.0$ and top-p 1.0, and evaluations use $\tau = 1.0$ and top-p 0.95.

For test-time reinforcement learning we use the same settings but halve the number of trainers and workers.

**Execution Feedback and Goedel-Prover for Repair (Sec. 4.2)**: We use temperature $\tau = 0.2$ and top-p 0.95 with a maximum prompt length of 8192 and a maximum generation length of 32768.

**Iterative Shortening (Sec. 4.3)**: For iterations 1 through 6, we use temperature $\tau = 1.0$ and top-p 0.95. We increase the temperature to $\tau = 1.2$ for iteration 7, and to $\tau = 1.5$ for iteration 8. We find that the higher temperatures in later iterations are helpful for increasing diversity with 1024 samples.

**Lean Base Model (Sec. B.1)**: We use an effective batch size of 512 (2 nodes, 8 H100/node, 32 gradient accumulation steps) and learning rate 1e-5 with 100 steps of warm-up starting from 1e-30. We train with a maximum sequence length of 8192 for 2000 steps.

**Proof Sketching (Sec. B.2)**: We use an effective batch size of 64 (2 nodes, 8 H100/node, 4 gradient accumulation steps) and learning rate 1e-5 with 100 steps of warm-up starting from 1e-30. We train with a maximum sequence length of 8192 for 50 steps. Evaluation is done with temperature $\tau = 0.8$ and top-p 0.95.

**Comparison with Leading Models (Sec. E)**: For our model and Qwen2.5-32B, we use $\tau = 1.0$ and top-p 0.95. For GPT-4o and Gemini-2.5-Pro, we use the default settings with $\tau = 1.0$.

## L PROMPTS

### L.1 PROOF SIMPLIFICATION PROMPT

```
You are given a correct Lean 4 proof of a mathematical theorem.
Your goal is to simplify and clean up the proof, making it shorter and more readable while
    ↪ ensuring it is still correct.

Here is the original proof:
```lean4
{statement}
```

Now, provide your simplified proof. Do NOT modify the theorem or header, and surround your
    ↪ proof in ```lean4 and ``` tags.
```

Listing 22: Zero-shot Proof Sketching Prompt

### L.2 PROOF SKETCHING PROMPTS

```
Your task is to translate a natural language math solution into a Lean 4 proof sketch that
    ↪ follows the structure of the natural language solution. Follow these guidelines:
1. Analyze the natural language solution and identify the key steps.
2. Translate each key step into Lean 4 syntax, structuring your proof using 'have' statements
    ↪ for clarity. Include all core steps from the natural language solution.
3. Use 'sorry' to replace individual proofs of lower-level steps, ensuring that your proof
    ↪ skeleton would compile successfully in Lean 4.
4. Surround your Lean 4 proof sketch in ```lean4 and ``` tags.

Problem:
{problem}

Solution:
{solution}

Lean 4 Statement:
```lean4
{statement}
```

Now, provide your Lean 4 proof sketch. Do NOT modify the theorem or header, and surround your
    ↪ proof sketch in ```lean4 and ``` tags.
```

Listing 23: Zero-shot Proof Sketching Prompt

```
Your task is to translate a natural language math solution into a Lean 4 proof sketch that
    ↪ follows the structure of the natural language solution. Follow these guidelines:
1. Analyze the natural language solution and identify the key steps.
2. Translate each key step into Lean 4 syntax, structuring your proof using 'have' statements
    ↪ for clarity. Include all core steps from the natural language solution.
3. Use 'sorry' to replace individual proofs of lower-level steps, ensuring that your proof
    ↪ skeleton would compile successfully in Lean 4.
4. Surround your Lean 4 proof sketch in ```lean4 and ``` tags.

Here is an example:

Problem:
Prove that if p, q are primes such that q is divisible by p, then p must be equal to q.

Solution:
Since q is prime, it only has 2 divisors: 1 and itself. Therefore, since p divides q, either
    ↪ $p=1$ or $p=q$. Because $p$ is a prime, $p \ne 1$, so $p=q$.

Lean 4 Statement:
```lean4
import Mathlib

theorem prime_divides_prime_equal (p q : ℕ) (hp : Prime p) (hq : Prime q) (h : p | q) : p = q
    ↪ := by sorry
```

Lean 4 Proof Sketch:
```lean4
import Mathlib

theorem prime_divides_prime_equal (p q : ℕ) (hp : Prime p) (hq : Prime q) (h : p | q) : p = q
    ↪ := by
```

```
  -- Lemma 1: Since q is prime, it only has 2 divisors: 1 and itself.
  have lemma1 : p = 1 ∨ p = q := by
    sorry

  -- Lemma 2: Since p is prime, p ≠ 1.
  have lemma2 : p ≠ 1 := by
    sorry

  -- Now, do case analysis on lemma1 to conclude p = q.
  cases lemma1 with
  | inl h_left =>
    contradiction
  | inr h_right =>
    exact h_right
```

Now, it is your turn to provide your Lean 4 proof sketch for a new problem. Do NOT modify the
    ↪ theorem or header, and surround your proof sketch in ```lean4 and ``` tags.

Problem:
{problem}

Solution:
{solution}

Lean 4 Statement:
```lean4
{statement}
```

Lean 4 Proof Sketch
```

Listing 24: One-shot Proof Sketching Prompt

### L.3 GOEDEL-PROVER REPAIR PROMPT

In Listing 25, use a modified version of Goedel-Prover's repair prompt found in their codebase. The
main difference is that because we do not have proofs annotated with CoT's, our `lean_proof` only
contains a proof.

```
Complete the following Lean 4 code:

```lean4
{formal_statement}```

Before producing the Lean 4 code to formally prove the given theorem, provide a detailed proof
    ↪ plan outlining the main proof steps and strategies.
The plan should highlight key ideas, intermediate lemmas, and proof structures that will guide
    ↪ the construction of the final formal proof.

Here is the proof:
```lean4
{lean_proof}```

The proof (Round 1) is not correct. Following is the compilation error message, where we use <
    ↪ error></error> to signal the position of the error.

{error_message_for_prev_round}

Before producing the Lean 4 code to formally prove the given theorem, provide a detailed
    ↪ analysis of the error message.
```

Listing 25: Goedel-Prover Repair Prompt

## M  PYTHON CODE FOR PROOF LENGTH

```python
import re
from collections import Counter

def proof_length(statement_and_proof):
    lean_operators = [':=', '!=', '&&', '-.', '->', '←', '..', '...',
    ↪ '::', ':>',
                      '<;>', ';;', '==', '||', '=>', '<=', '>=', '⁻¹',
    ↪ '?_']
    lean_operators_spaced = [' '.join(conn) for conn in lean_operators]
    lean_operators_dict = dict(zip(lean_operators_spaced,
    ↪ lean_operators))
    def lexer(lean_snippet):
        tokenized_lines = []
        for line in lean_snippet.splitlines():
            tokens = []
            token = ''
            for ch in line:
                if ch == ' ':
                    if token:
                        tokens.append(token)
                        token = ''
                elif str.isalnum(ch) or (ch in "_.'"):
                    token += ch
                else:
                    if token:
                        tokens.append(token)
                        token = ''
                    tokens.append(ch)
            if token:
                tokens.append(token)
            tokenized_line = ' '.join(tokens)
            for conn in lean_operators_spaced:
                if conn in tokenized_line:
                    tokenized_line = tokenized_line.replace(conn,
    ↪ lean_operators_dict[conn])
            tokenized_lines.append(tokenized_line)
        return '\n'.join(tokenized_lines)

    def remove_statement(statement_and_proof):
        if ":= by" in statement_and_proof:
            return statement_and_proof.split(":= by",
    ↪ maxsplit=1)[1].strip()
        return statement_and_proof.split(":=", maxsplit=1)[1].strip()

    def remove_comments(lean_snippet):
        # multi-line comments
        lean_snippet = re.sub(r" */-.*-/", "", lean_snippet,
    ↪ flags=re.DOTALL)
        # single-line comments
        lean_snippet = re.sub(r" *--.*", "", lean_snippet)
        return lean_snippet

    try:
        proof = remove_statement(statement_and_proof)
        proof = remove_comments(proof)
        proof_tokenized = lexer(proof)
        return sum([len(l.split(' ')) for l in
    ↪ proof_tokenized.splitlines()])
    except:
        return 10**9
```

