# OpenReview forum: "ProofOptimizer: Training Language Models to Simplify Proofs without Human Demonstrations"
_ICLR.cc/2026/Conference — ICLR 2026 Poster_

### Official Review · Reviewer_As7x · 2025-10-28

**Soundness:** 4
**Presentation:** 4
**Contribution:** 3
**Rating:** 8
**Confidence:** 4

**Summary:**

ProofOptimizer is a system that optimizes the length of long Lean proofs, such as those produced by SOTA prover models. ProofOptimizer is the first work to use reinforcement learning to tackle this problem setting, and they show by comprehensive experiments that they improve upon existing baseline, reaching substantial proof reduction.

**Strengths:**

ProofOptimizer is the first work to use RL to shorten formal proofs. The experiments in Section 4 conclusively show the advantage of this method over existing baselines such as Gemini-2.5-Pro on a pipeline like that of ImProver.

I am convinced by the analysis in Section 5 that optimized proofs are easier to learn for a model, and faster to compile. The former is important in improving the data quality in existing SFT datasets for Lean, and the latter is important considering the multi-day runs that AlphaProof, Seed-Prover, and Aristotle required for IMO problems. These establish the importance of the problem setting, which has not been analyzed in prior work.

**Weaknesses:**

The training distribution seems limited to high-school competition problems, which seems to limit the usability of ProofOptimizer for integrating long proofs of other types of results into libraries like Mathlib, as authors mention on L43–44.

**Questions:**

While proofs produced by Seed-Prover are very redundant and, to a human, look easy to simplify, I am very curious whether the ProofOptimizer pipeline can also be applied to human-written proofs like those in Mathlib, or helping verbose, model-written proofs (like those written by Aristotle) be merged into Mathlib, which is a key factor limiting such models. Relatedly, I am curious whether the model could do better by training on data outside Goedel-Pset or proofs written by Goedel-Prover-V2, such as human-written proofs or diffs in Mathlib commits. I recognize this may be a big ask and I don’t expect a complete answer in the author rebuttal stage.

Reduction @64x2 seems much better than @128 in Figure 3. Is there any further analysis on the tradeoff between iteration count and k? Relatedly, why the specific schedule of 6 iterations of k = 64 and 2 iterations with k = 1024?

I am interested in how the Lean-specific tokenizer works specifically, because it seems central to interpreting the numbers like min@k and for reproducibility. Or if the code is not long, is there code for the tokenizer? (You can probably put this as a small explanation in the Appendix).

---

> ### Author Response · Authors · 2025-11-21
>
> Dear Reviewer As7x,
> Thanks a lot for the detailed and constructive review of our work! We have uploaded a revised version, and below we address the weaknesses and questions you pose:
>
> ## Beyond high-school competition problems
> Beyond developing a strong proof optimization workflow, another goal of our work is to scientifically explore different design decisions and tradeoffs for both training and inference. For example, we show that training with RL leads to diversity collapse (Sec 4.1) and that using execution feedback is often less effective than increasing the sampling budget (Sec 4.2). While this work relies primarily on model-written proofs and competition-level problems, we expect our methodology and findings to generalize to the settings you mention.
>
> As a preliminary step, we add new experiments showing that the same iterative simplification scheme can also be applied for proof speedup instead of proof shortening by changing the target metric from proof length to heartbeat count (Sec 5.2.1 and Appendix I.1), leading to an inference pipeline for generating faster proofs.
> While addressing the concern of human-written proofs or domains like Mathlib is beyond the scope of this work, we believe these are promising directions and look forward to seeing the community explore these use cases as future work!
>
> ## Iteration count vs. k:
> In our opinion, reduction @64x2 is better than @128 because in the second iteration of @64, the model is already starting with a shorter proof, leading to more simplification opportunities compared to starting with a longer proof. In Appendix D.2, we analyze the metrics for different values of k and show how they differ for the initial iteration and a later iteration. Overall, as iteration count increases, proofs become harder to simplify in a low-sample setting.
>
> The schedule we choose is relatively arbitrary and balances experimentation speed with performant results. For example, we expect better results could be achieved using 128x16 rather than 2x1024), but this would require many more iterations. Cutting off at 8 iterations is also arbitrary, and running more iterations improves results even further.
>
> ## Lean-specific tokenizer
> Thanks for the interest! We have included the Python code for computing proof length in Appendix M (and you can run it on the proofs we attached in the supplementary material to reproduce our paper’s numbers).
>
> Again, thank you for all the comments! We kindly request that you reconsider your evaluation of our work in light of our responses, and we look forward to continuing the discussion and improving our work based on any further feedback you may have.

---

> > ### Comment · Reviewer_As7x · 2025-11-21
> >
> > Thank you for the response. I think this is a good paper worth presenting at ICLR. I am maintaining my current score of 8.

---

### Official Review · Reviewer_gPDN · 2025-10-30

**Soundness:** 3
**Presentation:** 3
**Contribution:** 3
**Rating:** 6
**Confidence:** 3

**Summary:**

This paper introduces ProofOptimizer, the first LLM designed to simplify formal proofs in the Lean theorem-proving system without relying on human-annotated simplification data. Addressing a critical bottleneck in neural theorem proving, where state-of-the-art (SoTA) reinforcement learning (RL)-trained provers (e.g., Seed-Prover, Goedel-Prover-V2) generate correct but excessively long, inscrutable proofs. ProofOptimizer integrates three core components: a symbolic Lean linter to remove redundant steps, a 7B-parameter LM fine-tuned for simplification, and an iterative inference-time workflow for progressive shortening. Trained via expert iteration (STaR-like iterative refinement using verified simplifications as training data) and online RL (rewarded for proof length reduction while preserving correctness), ProofOptimizer achieves promising results on miniF2F and PutnamBench with shortened proofs.

**Strengths:**

- **Address an important issue**: The paper targets proof simplification, which is an under-addressed problem in neural theorem proving. Unlike prior work (e.g., ImProver) that relies on agentic scaffolding around off-the-shelf LLMs and fails on long RL-generated proofs, ProofOptimizer is the first LLM trained specifically for simplification.
- **No Human Demonstrations**: By leveraging Lean’s mechanical verification to generate training signals (via expert iteration) and RL rewards (based on length and correctness), the model avoids the scarcity of human-annotated proof-simplification pairs, making its approach scalable and practical.
- **Strong Empirical Performance**: The results are compelling and well-validated across benchmarks.
- **Multiple Practical Benefits**: The paper goes beyond length reduction to demonstrate downstream value: simplified proofs speed up Lean execution and improve supervised fine-tuning of base provers.

**Weaknesses:**

- **Diversity Collapse in RL**: The paper acknowledges that RL training (ProofOptimizer-RL) improves single-sample performance (@1) but reduces multi-sample diversity (@32), limiting the model’s ability to explore alternative simplifications. This tradeoff is not fully resolved, and no mitigation strategies are proposed.
- **Computational Cost**: The iterative shortening workflow and RL training are computationally expensive: the authors note ~3000 H100 GPU hours per IMO problem. This high cost may limit adoption, especially for smaller research teams, and the paper does not discuss efficiency optimizations.
- **Dependence on Lean**: ProofOptimizer is designed exclusively for Lean, with no discussion of adaptability to other formal systems (e.g., Isabelle, Coq).

**Questions:**

See weaknesses.

---

> ### Author Response · Authors · 2025-11-21
>
> Dear reviewer gPDN,
>
> Thanks a lot for the detailed and constructive review of our work! We have uploaded a revised version, and below we address the weaknesses and questions you pose:
>
> ## Diversity collapse
> The goal of Sec 4.1 is to explore the trade-offs between different design choices in training, and therefore we explore three methods: expert iteration, RL, and test-time RL, finding that they excel in different scenarios. As you mention, for a single-shot setting, we would opt to use ProofOptimizer-RL (or test-time RL if there is further compute budget). However, where there is further inference budget (such as parallel sampling), we opt to use ProofOptimizer-ExpIt. This, for example, guided us to use ProofOptimizer-ExpIt (rather than RL) in the downstream iterative shortening in Sec 4.2 and 4.3.
>
> Therefore, we see this result as an interesting scientific finding, confirming previous mode collapse findings [1, 2, 3] in the proof optimization domain. Several mitigation strategies such as pass@k policy optimization [2] have recently been proposed to address the issue of diversity collapse in RL. We agree that they would be very interesting directions for exploration, but opt to use ProofOptimizer-ExpIt for achieving better multi-sample diversity.
>
> [1] https://arxiv.org/abs/2410.02089 \
> [2] https://arxiv.org/abs/2505.15201 \
> [3] https://arxiv.org/abs/2504.13837
>
> ## Computational cost:
> Our model is a 7B model, considered relatively small. DeepSeek-Prover-V2 is 671B, Seed-Prover has 20B active parameters (200B total), and Goedel-Prover-V2 is 32B (with a 8B variant). Yet, it outperforms larger models like Qwen2.5-32B, GPT-4o, and Gemini-2.5-Pro (Table 8).
> For RL training, each iteration of ProofOptimizer-ExpIt training only takes a few hours on 2 H100 nodes. Overall, the model requires under 300 GPU hours to train. If the goal is to shorten proofs, running inference and iterative shortening is even cheaper, even feasible on low-end GPUs and CPUs (albeit slower).
> Running one round of iterative shortening (with 64 samples) on our full batch of miniF2F problems (size 194) takes under an hour on a H100 node and already leads to a 77% reduction (see Table 5). With 6 iterations, we can achieve a 83% reduction.
> We used a large compute budget for our IMO experiments to explore the limits of our method rather than because it is necessary. Most of the progress occurs in the first few iterations: with just 200 GPU hours, we can already achieve a 30% average reduction. For more quantitative evidence, Appendix D.1 and D.2 show the proof reduction progress as sample size and iteration count increases.
>
> ## Dependence on Lean
> All recent SoTA LLM provers, as shown by the lack of leaderboard entries in Coq and Isabelle [1], are based on Lean. Our work relies on having access to these provers, which is only possible in Lean. That being said, our insights generalize to other formal systems.
> Beyond developing a proof optimization model, our work also scientifically explores different design decisions. For example, we show that training with RL leads to diversity collapse (Sec 4.1) and that using execution feedback is often less effective than increasing the sampling budget (Sec 4.2). These insights can be leveraged both for training provers in other languages and for optimizing for desiderata beyond proof length. To show the generality of our methodology, we add new experiments applying our iterative simplification scheme to generating faster proofs by changing the target metric (Sec 5.2.1 and Appendix I.1).
>
> [1] https://trishullab.github.io/PutnamBench/leaderboard.html
>
> Again, thank you for all the comments! We kindly request that you reconsider your evaluation of our work in light of our responses, and we look forward to continuing the discussion and improving our work based on any further feedback you may have.

---

### Official Review · Reviewer_vYF8 · 2025-10-31

**Soundness:** 4
**Presentation:** 4
**Contribution:** 2
**Rating:** 6
**Confidence:** 3

**Summary:**

This work introduces ProofOptimizer, a 7B LLM for Lean proof shortening trained from Qwen2.5-7B-Instruct through the following stages:
(1) Fine-tuning on five Lean-related tasks
(2) Expert Iteration (with a 0.8len(x) threshold as the filter) on the Proof Simplification dataset built by collecting and extracting theorems from Goedel-Pset and generating proofs with Goedel-Prover-V2-32B
(3) RL using a variant of GRPO with a reward $R(x, y) = max (|y|/|x| - 1, 0)$ for shorter proof $y$ given the prompt proof $x$.

When further Integrated with several inference-time techniques, ProofOptimizer manages to significantly reduce the length of the proofs generated by Goedel-Prover-V2-32B by 72.5% and 23.8% on miniF2F and PutnamBench, respectively. Besides, the ProofOptimizer can be used to optimize the training data for fine-tuning a theorem prover to achieve 2% performance gain.

**Strengths:**

- The proof shortening performance of ProofOptimizer is impressive. Seed-Prover’s IMO 2025 proofs are shortened by 50%, which makes the proof much easier for human to go through.

- The task and all the techniques used in the pipeline are clearly stated, and the results are presented in the easiest way for the readers to acquire and understand.

- The performance evaluation and ablation studies are comprehensive. Almost all the questions I came up during reading are answered in the paper.

**Weaknesses:**

1. The overall conceptual novelty of this work is limited. The task of shortening Lean proof can be seen as a code simplification task on the functional programming language Lean. The main contribution is on engineering implementation.

2. The significance of pursuing the "shortest proof" is not entirely clear. In the introduction, the authors claim that "RL-trained provers often generate proofs that are correct but excessively long and inscrutable", where I do not agreee with the last "inscrutable" description. Both Kimina-Prover and DeepSeek-Prover-V2 follow a declarative proof style, so do the subsequent Goedel-Prover-V2 and Seed-Prover. Compared to the inscrutable imperative(or procedural) style proofs such as the IMO-2024 solutions made by AlphaProver, declarative style proofs explicitly claim every intermediate goal in the proof, which are understandable for people who are familiar with the basic grammars of the formal language. According to Figure 11, a large part of the red@k improvement of ProofOptimizer comes from using a single tactic to substitute the whole proof, which could adversely reduce the readability of the proof.
In my opinion, the ideal proof rewriter should do more than shortening the proof. It should not only remove all reduntant tactics, but also make sure the proof granularity is not more coarse-grained than typical natural langauge proofs by human. In other words, advanced tactics that omit necessary details should not be used in the rewrite.

**Questions:**

1. What do you think is the "best" proof for a theorem in the formal system? In my opinion, since a single tactic can comprise complex constructions of proofs (such as brute force tactics mentioned in section 5.2) and can thus be time-costing, it seems to me that shorter execution time is a better metric for evaluating a proof than the proof length. Can you share your understanding on this topic?

2. For non-thinking models, we can directly shorten the proofs in the training data and fine-tune the model on the shortened data, as section 5.1 does. What shall we do when it comes to thinking models that only output the proof at the end of the response?

---

> ### Author Response · Authors · 2025-11-21
>
> Dear Reviewer vYF8,
>
> Thanks a lot for the detailed and constructive review of our work! We have uploaded a revised version, and below we address your comments:
>
> ## Conceptual novelty
> To our knowledge, there has only been a very limited body of work addressing code simplification. Formal theorem proving is a core application of simplifying LLM-generated code due to the convoluted proofs caused by RL on proof correctness signals, which is less the case for other code generation tasks.
> In addition to the engineering implementation, our paper highlights several novel analyses and insights. For example, we show the relative benefits of expert iteration/GRPO/test-time GRPO (Sec 4.1). We also analyze why our scheme of using execution feedback with Goedel-Prover is often less effective than increasing the sampling budget (Sec 4.2). We also show the benefit of training on shortened proofs (Sec 5.1), which has not been previously explored in the AI for formal math literature.
>
> ## Significance of proof shortening
> While we agree that declarative-style proofs can often be easier to read than imperative style proofs, we believe that even declarative proofs can be inscrutable. For example, despite knowing many Lean experts, we have not met anyone who was able to explain what is going on behind Seed-Prover’s 2025 IMO P1 [1]. Even though each intermediate goal in the proof is written, the reader can often lose track of the flow of the proof due to its length, nesting depth, and convoluted nature. There are 1,459 “have” statements in this particular proof, much more than would be in the natural language analogue.
>
> [1] github.com/ByteDance-Seed/Seed-Prover/blob/main/SeedProver/imo2025/IMO2025/P1.lean
>
> In Figure 11, the single-tactic phenomenon comes from the fact that miniF2F has many simple problems, such as showing (1 * 3 * 5 * 7 * 9 * 11 * 13) % 10 = 5 (mathd_numbertheory_299). While it is a matter of taste, we believe it is justified to use advanced tactics to abstract away details, similar to how some simpler lemmas in math textbooks are “left to the reader”. If desired, Lean allows for the possibility of elaborating advanced tactics by more detailed tactic sequences.
>
> ## “Best” proof
> Thanks for the very interesting question! Like with programming, proof-writing requires a balance of many desires including correctness, abstraction design, readability, and performance. We believe there is no universal definition of the “best” proof, as different people have varied tastes [1, 2, 3]. In fact, the Lean forum often has extended discussions over the correct way to structure a proof.
> As mentioned in L106-115 (revised version), our approach is metric-agnostic, and proofs can be optimized for length, execution time, or other metrics including those in [3]. We choose proof length for its simplicity and ease of measurement: after all, Mathlib has over 2k pull requests golfing proofs [4].
> To demonstrate metric-agnosticity, we add new experiments changing the target metric to Lean heartbeats (Sec 5.2.1 and Appendix I.1), leading to a pipeline for generating proofs with a shorter execution time. An interesting finding is that proof length and proof execution time are actually highly correlated, and that optimizing for one often leads to gains in the other for free!
>
> [1] https://gowers.wordpress.com/2025/09/22/creating-a-database-of-motivated-proofs \
> [2] https://arxiv.org/abs/2405.04699 \
> [3] https://arxiv.org/abs/2410.04753 \
> [4] https://github.com/search?q=repo%3Aleanprover-community%2Fmathlib4+golf&type=pullrequests
>
> ## Thinking models
> The same paradigm can be applied to thinking models. When doing expert iteration on a thinking model, the generated data will consist of (original proof, thinking chain, simplified proof) triples instead of just (original proof, simplified proof) pairs. Another enhancement could be to generate synthetic rationale chains CoT data from question/answer pairs such as in [1, 2], which would allow distilling a stronger model’s reasoning chains into a weaker initial model to better bootstrap the first few rounds of expert iteration.
>
> [1] https://arxiv.org/abs/2506.08388 \
> [2] https://arxiv.org/abs/2309.05653
>
> Again, thank you for all the comments! We kindly request that you reconsider your evaluation of our work’s contribution in light of our responses, and we look forward to continuing the discussion and improving our work based on any further feedback you may have.

---

### Official Review · Reviewer_524q · 2025-11-04

**Soundness:** 3
**Presentation:** 3
**Contribution:** 3
**Rating:** 6
**Confidence:** 3

**Summary:**

The goal of this work is to shorten LLM generated proofs, which can be long and difficult to understand. This work trains a new LM to shorten a given proof, using reinforcement learning. The reward signal is a mixture of the Lean correctness reward and the length of the new proof output by the model.

First, this work introduces a Lean linter, which removes some obviously inefficient portions of proofs. Afterwards, to train an LM to shorten proofs, they use expert iteration, or RL. The initial model is Qwen-2.5-7B-Instruct, which is then further trained on mathematical/formal theorem proving-related tasks before expert iteration/RL. The training dataset is constructed using some problems from Goedel-Pset. The problems’ natural language solution is turned into a proof sketch. Then, each step in the proof sketch is used as a theorem (and the solution is given by Goedel-Prover-V2-32B), resulting in 145K theorem-proof pairs.

Let M be the proof-shortening model. Within each round of expert iteration, for each theorem-proof pair, several shorter versions of the proof are sampled from M, and the shortest correct proof is included in the SFT dataset within this round of expert iteration. The length of this proof should be less than 80% the length of the original proof, so that the model will learn some non-trivial simplifications. In reinforcement learning, the reward is the relative shortening (e.g. 0.1 if the proof is 10% shorter) if the proof is correct and 0 otherwise. The algorithm is GRPO, with no advantage normalization.

After performing 3 rounds of expert iteration (obtaining ProofOptimizer-ExpIt), the proof length with 1 generation, and minimum proof length among 32 generations, both improve significantly. However, when performing RL starting from 2 rounds of expert iteration, min@1 improves compared to ProofOptimizer-ExpIt, but the min@32 is worse. Their proof shortening models (both with expert iteration and RL) significantly outperform Gemini-2.5-Pro.

Additionally, this work finds that iteratively shortening the proof using repeated applications of their model - in each round, they generate several proofs and take the shortest one. There is a 57.2% proof length reduction on Putnam Bench, though some of the longer proofs are more difficult to simplify. There is a significant improvement from using more samples per iteration. Using a large sampling budget, there is around 40-50% length reduction on Seed Prover’s IMO proofs, using iterative applications of the proof-shortening model trained with expert iteration.

This work also finds that training on simplified proofs can help obtain better theorem proving models - when training their base model on proofs generated by Goedel-Prover-V2, they find that applying the proof-shortening model to these proofs before SFT can improve pass@32 on miniF2F. An additional benefit is that these proofs can execute/be verified more quickly.

**Strengths:**

1. Results with expert iteration are strong, as well as with iterative shortening of proofs using the expert iteration model.

**Weaknesses:**

1. The @1 metrics with RL are strong, but the @32 metrics with RL are not much better than the @1. So this may be a disadvantage since the @32 metrics also tend to be important (since it is possible to use parallel sampling together with the Lean verifier).

**Questions:**

1. Line 158 - is there a typo in the definition of relative shortening, i.e. should be (|x| - |y|)/|x| instead?
2. Why do you believe that min@32 is not significantly different from min@1 with RL? Is this due to an entropy collapse?

---

> ### Author Response · Authors · 2025-11-21
>
> Dear Reviewer 524q,
>
> Thanks a lot for the detailed and constructive review of our work! We have uploaded a revised version, and below we address your comments:
>
> ## @1 and @32 metrics; entropy collapse
> Different methods have different @1 vs @32 tradeoffs, as shown in Table 1 and explored in Sec 4.1. ProofOptimizer-RL should only be used in the single-sample setting. If there is more test-time compute budget, ProofOptimizer-ExpIt (which has better @32 metrics) should be used instead of ProofOptimizer-RL, as shown by the success of iterative simplification (Sec 4.3).
>
> Yes, we believe the @1 vs @32 gap is due to entropy collapse, as discussed in L218-224 (revised). This phenomenon has also been shown in code [1] and informal math [3]. An analysis of why is given in [2]: GRPO samples multiple attempts but rewards only the best, optimizing for pass@1 performance.
>
> [1] https://arxiv.org/abs/2410.02089 \
> [2] https://arxiv.org/abs/2505.15201 \
> [3] https://arxiv.org/abs/2504.13837
>
> ## Typos
> Thanks! We have corrected this in the revised version.
>
> Again, thank you for all the comments! We kindly request that you reconsider your evaluation of our work in light of our responses, and we look forward to continuing the discussion and improving our work based on any further feedback you may have.

---

### Author Response · Authors · 2025-11-21
**Shared Response to All Reviewers**

Dear Reviewers,

Thanks to all of you for the insightful reviews, suggestions, and questions on our work. We have uploaded a revised version (updates in green). We would like to re-emphasize that our work is metric-agnostic. To show this, we have added a new experiment running our iterative simplification pipeline using a proof execution time metric (Sec 5.2.1 and Appendix I.1). We have also included code to compute proof length in Appendix M, as requested by Reviewer As7x.

Overall, if our responses addressed your concerns, we hope that you update your evaluation of our work. Otherwise, we look forward to further questions and continued discussion.

Best,
Authors

---

### Meta-Review · Area_Chair_tC7B · 2026-01-07

**Summary:**

Here is a summary of the principal reviewers' concerns

 - Objective misalignment. Optimizing token-length shortest-ness can sacrifice human readability and often collapses to opaque “single-tactic” proofs. (Reviewer 524q, vYF8)
 - RL mode/diversity collapse. RL boosts @1 but hurts @32 (parallel-sampling regime), suggesting entropy collapse and limiting practical gains. (Reviewer 524q, gPDN)
 - Practicality and scope limits. very high compute for best-case demos, trained/evaluated mainly on competition-style, Lean-only, model-generated proofs, with unclear transfer to Mathlib/human proofs or other systems. (Reviewer gPDN, As7x)
 - Contribution/novelty skepticism. Viewed as largely engineering/code-simplification rather than a conceptually new ML contribution. (Reviewer vYF8)

**Reviewer Concerns:**

Here is how the reviewers' concerns were addressed by the rebuttal.

Reviewer 524q
- @1 vs @32 under RL  and diversity collapse.
Only partially addressed. Authors explicitly frame ProofOptimizer-RL as single-sample and recommend ProofOptimizer-ExpIt / iterative shortening when you have parallel sampling budget, they also explicitly attribute the gap to entropy/diversity collapse and add discussion.
But the underlying issue is not fixed, RL still underperforms in multi-sample regimes.

Reviewer vYF8
 - Limited conceptual novelty (engineering-heavy).
Largely unaddressed. Authors argue (i) little prior “code simplification” work, (ii) formal proving is a compelling domain, and (iii) the paper contributes analyses/insights. This is largely a judgment call, the rebuttal may not have changed a reviewer’s view if they still see it as primarily engineering.
 - Significance of “shortest proof” + readability risk from single-tactic collapse.
Partially addressed. Authors defend that even declarative proofs can be “inscrutable” due to sheer length/nesting; they justify advanced tactics as legitimate abstraction and note tactics can be elaborated if desired. They also claim metric-agnosticity and add experiments optimizing heartbeats (runtime proxy).The key worry remains, shorter can be less interpretable, and the work doesn’t enforce a readability/granularity constraint (it defends the choice rather than guaranteeing human-friendly outputs).
 - Thinking-models question.
Partially addressed. Authors propose training on triples (orig proof, CoT, simplified proof) and/or synthetic rationale distillation.
No demonstrated results, it’s a proposal, not evidence.

Reviewer gPDN
 - RL diversity collapse not mitigated.
Outstanding. Authors acknowledge the tradeoff, cite recent mitigation directions, but recommend using ExpIt for multi-sample diversity rather than solving collapse inside RL here. A concrete mitigation strategy + results are still missing.
 - Compute cost (~3000 H100 hours/IMO).
Addressed. Authors argue training cost is modest (7B; <300 GPU-hours), that the huge IMO budget was to probe limits, and that meaningful shortening occurs with far fewer iterations/samples. They provide runtime examples on miniF2F.
Although the paper may still lack a crisp efficiency-optimized recipe or “cost vs gain” guidance that convinces smaller teams.
 - Lean-only dependence / portability.
Essentially unaddressed. Authors argue Lean is where SOTA provers/benchmarks are, add metric-swap experiment as evidence of generality. But provide no direct experiments on Coq/Isabelle, portability remains unproven.

Reviewer As7x
- Training distribution narrow (competition problems) / Mathlib relevance.
Outstanding. The central request, evidence on human-written Mathlib proofs / broader domains, has not been addressed empirically.
 - Iteration schedule vs k tradeoff (why 64\times 6 then 1024\times 2).
Addressed. Authors give an intuitive reason why iterating can beat a single higher-k round (starting from a shorter proof opens more simplifications) and say the schedule is somewhat arbitrary, point to appendix analysis.
 - Tokenizer / reproducibility.
Partially addressed. Authors  included code for computing proof length in an appendix, but there is still no access to  the paper source code.

**Reviewer Scores:**

Because of many concerns  only partially addressed or remained unaddressed, the overall scores would essentially remain the same. Also during the normal AC-reviewers exchange, the reproducibility concern, i.e. the absence of access to the paper source code, would have likely put the downward pressure on the scores. The acceptance recommendation is contingent on the authors  providing a link to the paper source code for the camera ready revision.

---

### Decision · Program_Chairs · 2026-01-26

Accept (Poster)